# MedLesionVQA: A Multimodal Benchmark Emulating Clinical Visual Diagnosis for Body Surface Health

**Deli Yu**[1][*][‡]**, Shengzhi Wang**[1][*]**, Kai Wu**[1][*]**, Xiaozhong Ji**[1][*]**, Bo Cui**[1][*]**, Jieqiong Cao**[1][*]**,
Huichao Wang**[1][*]**, Boyuan Jiang**[1]**, Xu Wang**[1]**, Qian Xu**[1]**, Chao Gao**[1]**, Dan Wang**[1]**, Yi Zhao**[2]**,
Dian Chen**[2]**, Meng Li**[1]**, Haifeng Wu**[1]**, Yijun He**[1]**, Haihua Yang**[1][†]

[1]Xiaohe Team, Bytedance
[2]Dermatology, Beijing Tsinghua Changgung Hospital

`dely-yu@qq.com, wukaiwork@gmail.com, yanghaihua@bytedance.com`

## Abstract

Body-surface health conditions, spanning diverse clinical departments, represent some of the most frequent diagnostic scenarios and a primary target for medical multimodal large language models (MLLMs). Yet existing medical benchmarks are either built from publicly available sources with limited expert curation or focus narrowly on disease classification, failing to reflect the stepwise recognition and reasoning processes physicians follow in real practice. To address this gap, we introduce MedLesionVQA, the first benchmark explicitly designed to evaluate MLLMs on the visual diagnostic workflow for body-surface conditions in large scale. All questions are derived from authentic clinical visual diagnosis scenarios and verified by medical experts with over 20 years of experience, while the data are drawn from 10k+ real patient visits, ensuring authenticity, clinical reality and diversity. MedLesionVQA consists of 12K in-house images (*never publicly leaked*) and 19K expert-verified question–answer pairs, with fine-grained annotations of 94 lesion types, 110 body regions, and 96 diseases. We evaluate 20+ state-of-the-art MLLMs against human physicians: the best model reaches 56.2% accuracy, far below primary physicians (61.4%) and senior specialists (73.2%). These results expose the persistent gap between MLLMs and clinical expertise, underscoring the need for the multimodal benchmarks to drive trustworthy medical AI. The dataset can be found in `https://github.com/bytedance/MedLesionVQA`.

## 1 Introduction

Photo-based interaction with multimodal large language models has recently gained attention as a potential pathway for addressing body-surface health concerns, including the skin, nails, hair, oral cavity, genitals, and other visible areas(AlSaad et al., 2024; Zhou et al., 2024a). It requires MLLMs (Saab et al., 2024; Moor et al., 2023; Li et al., 2023a; Chen et al., 2024b; Lin et al., 2025; Nath et al., 2024) and medical MLLMs (Tian et al., 2023; Chen et al., 2023; Wei Zhu & Wang, 2023; Wang et al., 2025) to give visual diagnosis results according to body lesion images photographed by users via smartphone or other device. Although current MLLMs have shown the ability for medical assistance (Esteva et al., 2017; Coustasse et al., 2019; Tschandl et al., 2020), they still struggle to replicate the visual diagnostic workflow (Weller et al., 2014) that physicians rely on for body-surface health—spanning fine-grained recognition, reasoning, diagnosis, and treatment suggestions across departments such as dermatology, dentistry, and general surgery. The critical challenge is how to evaluate whether MLLMs can truly align with this workflow and perform like physicians in authentic clinical settings.

---

* Equal contribution
‡ Work done during internship
† Corresponding author

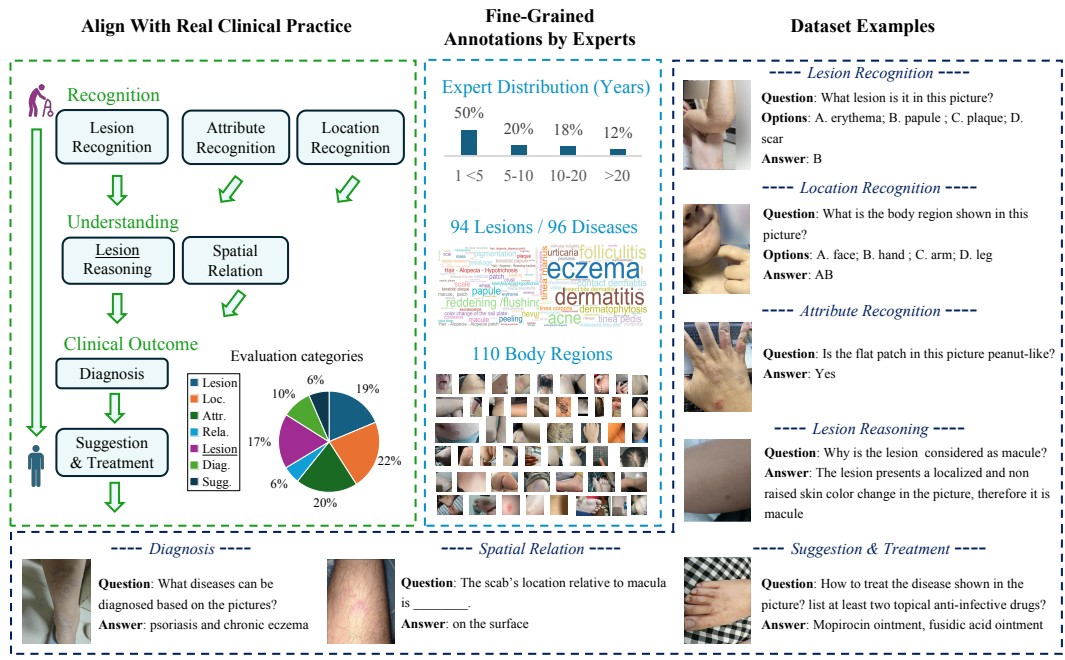

Figure 1: Overview of MedLesionVQA. The benchmark is designed to emulate the visual diagnostic workflow of physicians (top-left), covering seven core abilities with fine-grained annotations. Expert physicians with over 20 years of experience validated annotations (middle), which include detailed identification of 94 lesion types, 96 diseases, and 110 body regions (bottom).

Existing medical benchmarks are either assembled from publicly available sources with limited expert curation or focus narrowly on disease classification, failing to capture the visual diagnostic workflow for body-surface health that physicians follow in practice. General-purpose benchmarks, such as GMAI-MMBench (Ye et al., 2024) and OmniMedVQA (Hu et al., 2024), extend to up to 38 modalities by aggregating data from open-source websites. Although these datasets are extensive, publicly sourced information often includes outdated or basic-level data and lacks expert annotations critical for lesion interpretation and treatment recommendations. Conversely, specialized datasets such as SkinCon (Daneshjou et al., 2022b) and DDI (Daneshjou et al., 2022a) integrate expert annotations but focus narrowly on singular tasks, such as disease classification, not adequately reflecting real-world clinical practice. For instance, SkinCon (Daneshjou et al., 2022b) introduces lesion concepts, which are visual symptoms of disease, without open-ended diagnostic queries. DDI employs binary labeling (e.g., malignant vs. benign), which oversimplifies the real-world clinical complexities. Additionally, SkinCon contains only 3,700 images, and DDI encompasses merely 656 cases (Daneshjou et al., 2022a), which are insufficient for robust evaluation.

To address these issues, we introduce MedLesionVQA, the first benchmark explicitly designed to evaluate the visual diagnostic workflow for body-surface health. To ensure authenticity and close alignment with physician practice, we collaborated with senior medical directors with over 20 years of experience and defined seven core diagnostic abilities by referring to authoritative textbooks and clinical literature (Weller et al., 2014). These abilities span lesion recognition, reasoning, diagnosis, and treatment across dermatology & STD, dentistry, and surgery. Our dataset comprises 12K images collected directly from real patients, guaranteeing that *none originate from internet sources or leaked repositories*. With 12K images and 19K question–answer pairs, MedLesionVQA is substantially larger than prior expert-curated benchmarks for body-surface health, enabling more robust and diverse evaluation. Beyond its authenticity and scale, MedLesionVQA implements a fine-grained annotation system, covering 94 lesion types, 96 diseases, and 110 anatomical regions. For example, a human hand is subdivided into nine distinct regions, from the purlicue to the fingertip, enabling highly detailed evaluation of model performance.

Furthermore, our QA generation pipeline is grounded in real clinical questions, which serve as templates for automatic generation and are then refined through rigorous expert review. This yields over 19K diverse, high-quality QA pairs with expert-level accuracy and statistical reliability, addressing gaps left by prior benchmarks. After extensive prompt tuning and iterative refinement, we establish

an LLM-based scoring system developed with physicians, ensuring strong consistency between automated assessments and human judgments. Our key contributions are summarized as follows:

- **The first body-surface benchmark aligned with visual diagnostic workflow.** We introduce the first multimodal benchmark explicitly designed to evaluate the visual diagnostic workflow for body-surface health, moving beyond narrow disease classification. MedLesionVQA evaluates the stepwise diagnostic abilities of state-of-the-art MLLMs, providing a foundation for their advancement toward real-world clinical use.
- **Expert-level and fine-grained annotation system.** Our benchmark benefits from valuable expert annotations, covering over 96 prevalent diseases, 110 body regions and sub-regions, and 94 distinct lesion types. All annotations are conducted and rigorously verified by clinical experts following a systematic clinical lexicon tree.
- **Comprehensive evaluation.** We conducted an extensive evaluation involving more than 20 widely-used MLLMs. Additionally, we established human baselines by engaging general practitioners and senior physicians, enabling a thorough and systematic comparison between MLLMs and medical experts.

## 2 RELATED WORKS

### 2.1 MULTIMODAL LARGE LANGUAGE MODELS

Numerous Multimodal Large Language Models have been developed, focusing primarily on improving image captioning, visual question answering, and cross-modal retrieval (Achiam et al., 2023; Anthropic, 2025a; Bai et al., 2023; Chen et al., 2024e;f; Liu et al., 2023c; Chen et al., 2024f;c). Representative models include the GPT-4V (Achiam et al., 2023), DeepSeek series (Guo et al., 2025), LLAVA series (Li et al., 2024; Liu et al., 2023c), InternVL series (Chen et al., 2024f;d), Qwen series (Bai et al., 2025; Wang et al., 2024b), and CogVLM series (Wang et al., 2024c; Hong et al., 2024), among others (Laurençon et al., 2023; Ding et al., 2021). These works have significantly contributed to the development of the community. To address specific medical tasks, researchers have trained and fine-tuned MLLMs using specialized medical data, leading to the development of medical vision-language models (Li et al., 2023a; He et al., 2024; Wu et al., 2023; Liu et al., 2023d), which integrate medical images (such as X-rays, MRIs, and CT scans, *etc.*) with clinical data (including patient records, diagnosis, and treatment plans, *etc.*) (Ye et al., 2024; Antonelli et al., 2022; Irvin et al., 2019). However, achieving precise medical question answering and fine-grained multimodal diagnostics remains a significant challenge.

Table 1: Difference between MedLesionVQA and other existing benchmarks. OmniMedVQA* (Hu et al., 2024) and GMAI-MMBench*(Ye et al., 2024) contains a subset of lesion images for dermatology-related evaluation.

| Benchmark | Images/QA | VQA | Data source | **Anno./Eval. dimension** |
|---|---|---|---|---|
| OmniMedVQA* (Hu et al., 2024) | 119K / 128K | ✓ | public | lesion (*unknown*)
body region (25) |
| GMAI-MMBench*(Ye et al., 2024) | 26K / 26K | ✓ | public | disease (*unknown*) |
| Fitzpatrick17K (Groh et al., 2021) | 17K / *null* | ✗ | public | disease (114) |
| DermNet (der, 2023) | 19K / *null* | ✗ | public | disease (23) |
| SkinCon (Daneshjou et al., 2022b) | 3230 / *null* | ✗ | public | lesion concepts (48) |
| DDI (Daneshjou et al., 2022a) | 656 / *null* | ✗ | in-house | disease (2) |
| SNU-134 (Han, 2019) | 2101 / *null* | ✗ | in-house | disease (134) |
| **MedLesionVQA** | 12K / 19K | ✓ | in-house | lesion (94) and attribute (7)
body region (110)
disease (96)
suggestion & treatment |

### 2.2 BENCHMARKS

The field of MLLMs has experienced rapid advancements, both in terms of models (Achiam et al., 2023; Bai et al., 2023; Anthropic, 2025a) and benchmarks (Bitton et al., 2023; Zhu et al., 2024; Li et al., 2025; Ray et al., 2024; Lim et al., 2024; Yu et al., 2023; 2024; Xu et al., 2023; Lee et al., 2024;

Roberts et al., 2024). Evaluating the medical capabilities of MLLMs requires specific benchmarks, and the representative medical benchmarks include VQA-RAD (Lau et al., 2018), SkinCon (Daneshjou et al., 2022b), SkinCAP (Zhou et al., 2024b), DDI (Daneshjou et al., 2022a), SCIN (Ward et al., 2024), SLAKE (Liu et al., 2021), RadBench (Wright & Reeves, 2016), MMMU (Yue et al., 2024), GMAI-MMBench (Ye et al., 2024), OmniMedVQA (Hu et al., 2024) and MediConfusion (Sepehri et al., 2024), *etc.*. Among which, OmniMedVQA (Hu et al., 2024) introduces the largest medical VQA dataset to date, covering 12 data modalities and 20 anatomical regions, with over 100k images. GMAI-MMBench (Ye et al., 2024) includes various medical imaging data, such as X-rays, CT scans, MRIs, and ultrasounds, along with corresponding clinical information. RadBench (Wright & Reeves, 2016) focuses on radiology, involving tasks such as modality recognition and disease diagnosis. DermaVQA(Yim et al., 2024) introduces a multilingual dermatology VQA dataset built from remote-care patient portal messages and focusing on response generation for dermatology consultations. Derm1M (Yan et al., 2025), in contrast, is a million-scale dermatology vision–language training dataset aligned with a clinical ontology. In this work, we introduce MedLesionVQA, which consists of 12K+ in-house body lesion images and 19K expert-verified QA pairs. It uniquely targets the stepwise visual diagnostic multimodal abilities that are central to real visual diagnosis workflows.

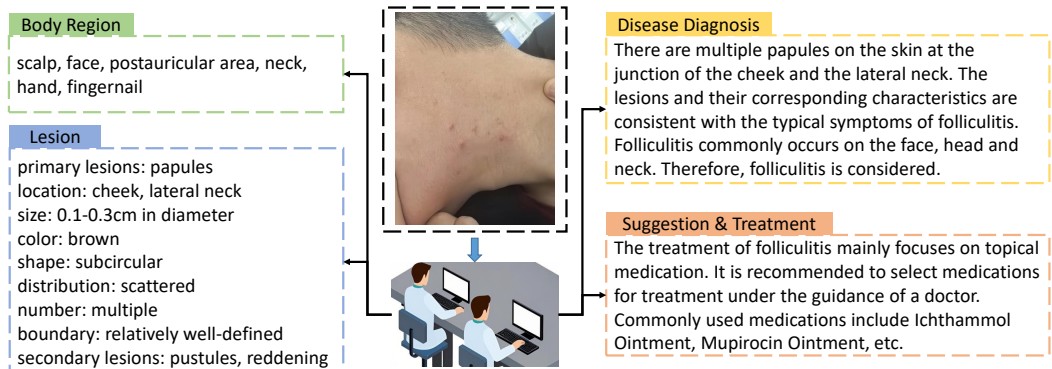

Figure 2: The Annotation procedure. The physicians sequentially annotate the body regions, lesions, attributes, disease diagnosis, and finally suggestion & treatments.

# 3 ESTABLISHMENT OF MEDLESIONVQA

## 3.1 OVERVIEW OF BENCHMARK

MedLesionVQA comprises 12K in-house patient images collected under approved ethical protocols. All images undergo a rigorous pre-processing pipeline, including image quality filtering, content inspection, de-identification of personal information, and distribution balancing. We cooperate with senior physicians to design and implement an annotation protocol, referencing authoritative materials (Weller et al., 2014; James et al., 2011). The protocol covers 96 prevalent diseases, 94 lesion types, and 119 body regions. Then, inspired by diagnosis and treatment pipeline in clinical practice, we construct 19K diverse question-answer samples involved with 7 stepwise visual diagnostic abilities, and some examples are shown in Fig. 1. Finally, we propose an automated scoring pipeline to calculate the metric of MLLMs' benchmark results, and the scoring pipeline is tuned to align physician judgment metric with negligible difference.

## 3.2 ANNOTATION PROTOCOL

More than tens of physicians are invited into the image annotation process, which contains image filtering, annotation labeling, and annotation reviewing. First, a group of annotators check the quality of each image, such as its clarity, and discard the unqualified images as well as those that do not show the exposed human skin or the oral cavity. Second, body region type, lesion type, lesion attribute type, disease type, and suggestion & treatment annotations are labeled under annotation rules, which are developed by an expert panel of senior experts. Finally, other senior experts review the annotation results and correct any errors, ensuring the annotation quality with entity-level precision and recall of over 95%.

**Body region.** The physicians are asked to annotate all visible parts of the human body and the internal parts of the oral cavity. We have respectively constructed the corresponding lexical trees for part division, and the annotation is carried out according to the secondary nodes of the lexical trees. More information of the lexical trees is detailed in Appendix A.2.

**Lesion.** Our dataset has annotations for 94 types of lesions. For each lesion, we describe its key attributes. These attributes are: size, color, shape, quantity, distribution, and boundary. We also pinpoint the exact location of each lesion. To do this, we use a very detailed body map, much like the fine branches of a tree. All our labels have multiple options, not just "yes or no," and most come with at least 7 different text descriptions. Finally, we identify primary and secondary lesions. We also describe their relationship and how often they appear together.

**Disease.** Each image is provided with up to 3 differential disease diagnosis by two independent physicians, which are sorted in the order they consider the most reasonable. Then, the inverse of the rank is used as the weight to combine the annotation results of the two physicians, to obtain the final sorting result. For the list of total disease labels in the annotation data, please refer to Table 4 of the supplementary material. The logic of diagnostic reasoning is also provided during annotation.

**Suggestion & Treatment.** For each image, physicians are required to provide corresponding treatment suggestions based on the unique disease diagnosis or differential disease diagnosis, including advice on seeking medical treatment, medication, matters needing attention in daily life, and so on.

### 3.3 QUESTION-ANSWER CONSTRUCTION

This section introduces the process of question generation, including category balance, prompt design tailored for assessing different cognitive abilities, and the development of various question types.

**Evaluation category balance.** We balanced the distribution of questions across seven abilities to closely reflect their real-world distribution in clinical practice, as illustrated in Fig. 1. Lesion, attribute, and location recognition questions comprise 61% of the MedLesionVQA dataset, as accurate fine-grained recognition is fundamental for subsequent diagnostic tasks. Specifically, the evaluation assigned equal weighting to each lesion type according to the real-world distribution, ensuring comprehensive coverage for accurate skin lesion identification and analysis.

**QA construction prompts.** In the context of real-world question examples, we design different QA generation templates for different evaluated abilities in order to test the corresponding capabilities. Two typical prompts are displayed in Fig. 3(a), and the rest will be included in the supplementary materials.

**Diverse question types.** The generated questions are categorized into two types: multi-choice and open-ended questions, while open-ended questions include judgment, fill-in-the-blank, and short-answer questions. For multi-choice questions, we create similar distracted options based on the correct answer and then randomize the order of all options, ensuring that the correct answer has an equal likelihood of appearing in any position. To prevent answers from being overly diverse and difficult to assess, the answers to open-ended questions are kept relatively concise. This approach enables the judging model to provide more consistent scores in the subsequent evaluation.

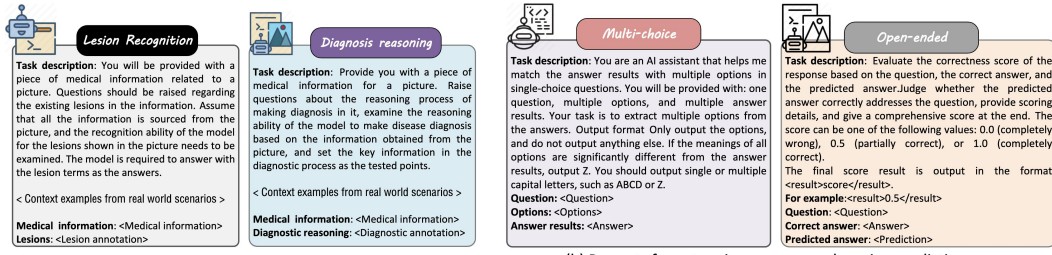

(a) Prompts of automatic QA construction for evaluation abilities.  (b) Prompts for extracting answers and scoring predictions.

Figure 3: The prompt template used on MedLesionVQ. Medical information includes body region, lesion, attribute, disease diagnosis, and suggestion & treatment information annotated above.

**Manual review and improvement.** To enhance the medical accuracy and ensure appropriate difficulty in QA sets, physicians manually review all auto-generated QA pairs. This review focuses primarily

on verifying the correctness of critical medical information within both the questions and the answers. Ambiguous questions are clarified, and non-standard answers are revised accordingly. Additionally, distractors in multi-choice questions are assessed regarding their accuracy and difficulty. A few open-ended questions, particularly those concerning suggestion & treatment and lesion reasoning, are converted into multi-choice format due to the inherent complexity of determining definitive answers. The final benchmark comprises 19,843 question-answer pairs (QAs), which are partitioned into a validation subset containing 1,499 QAs (7.55% of total samples) and a test subset consisting of 18,344 QAs (92.45% of total samples).

### 3.4 AUTOMATIC SCORING PIPELINE

For *multiple-response questions*, since MLLMs occasionally fail to output exact option answer, we need to extract the option answer from the answer set and the raw prediction output using extracting-answer prompt and then compare it with the correct answer. To calculate score, we have set the following rules: 1) If the predicted answer contains options that are not in the correct answer set, it is considered completely wrong and receives a score of 0; 2) If the predicted answer fails to identify all correct answers, the score is calculated based on the ratio of the number of correctly answered options to the total number of correct answers.

For *open-ended questions*, the prompt for the judge model is designed as indicated in Fig. 3(b). With this prompt, the judge model will analyze the predicted answer, compare its similarity to the correct answer, and most importantly, determine whether the question has been answered.

**Evaluation consistency test.** We use GPT-4 as judger to score the model's predicted answers for open-ended QAs. Moreover, we invite physicians to score the answers, also using the three scoring levels of $0 - 0.5 - 1.0$. Through the analysis of inconsistent cases, we find that the model is *too strict* in scoring for attributes such as color and size. For example, or color descriptions like "pink" and "skin tone", and size descriptions like "pinpoint" and "millimeter", due to the lack of specialized medical knowledge, the judge model tends to be overly strict according to general criteria. When we supplement the evaluation details for color and size in the prompt, therefore the high consistency rate between the judge model's scores and manual scores can be ensured. The details can be found in Appendix A.5.

## 4 EXPERIMENTS

### 4.1 EVALUATION

**MLLMs baseline.** For closed-source models, we evaluate several well-known models, including GPT series models (Achiam et al., 2023), Gemini series models (Google, 2025; DeepMind, 2024), and Claude4-opus(Anthropic, 2025a). For open-source models, we comprehensively evaluate model parameters ranging from 0.256 billion to 72 billion, including the famous LLaVA series(Liu et al., 2023b; Li et al., 2023b), Qwen2.5 series (Wang et al., 2024a), InternVL series(Chen et al., 2024f), DeepSeek-VL series (Wu et al., 2024) and several medical-domain models, like MedGemma (Sellergren et al., 2025), HuatuoGPT-vision (Chen et al., 2024b), Bimedix2 (Mullappilly et al., 2024), LLava-med (Li et al., 2023b), lingshu-7b (Xu et al., 2025), and lingshu-32b(Xu et al., 2025).

**Physician baseline.** We invite two groups of 15 primary and 15 senior physicians to answer the 1499 questions in the validation set, respectively. Primary physicians are general practitioner, while senior physicians are specialized expert from dermatology or dentistry departments. Questions are randomly distributed, and each question is completed by at least 2 different physicians. The physicians are not allowed to consult textbooks or search the Internet during the question completion task.

**Evaluation Implementation.** The evaluation is conducted using the VLMEvalKit (Duan et al., 2024) framework under a zero-shot setting. There are text-only set and vqa-set. We additionally add a text-only baseline input to isolate the contribution of the visual modality, helping to evaluate the model's reliance on visual versus textual information.

MedLesionVQA has around 9% questions, which can be answered without significant reliance on visual input. These questions mainly belong to the category of treatment and suggestion. We intentionally retained all questions rather than dropping the text-only ones out, since both visual and

Table 2: The overall accuracy of open-source and closed-source models on the test set and validation set. *:Some closed-source commercial models are evaluated only on the valid set due to API access limitations. The table is sorted in descending order based on the AVG_test score.

| Model | AVG_val (1499) | AVG_test (18344) | Recognition | | | Understanding | | | |
|---|---|---|---|---|---|---|---|---|---|
| | | | Lesion Recognition (3340) | Location Recognition (3986) | Attribute Recognition (3508) | Spatial Relation (1133) | Lesion Reasoning (3071) | Disease Diagnosis (1693) | Suggestion Treatment (1613) |
| Text + Image as Input | | | | | | | | | |
| Senior physicians* | 0.7321 | - | 0.6826 | 0.7583 | 0.7046 | 0.7102 | 0.6533 | 0.7313 | 0.8574 |
| Primary physicians* | 0.6144 | - | 0.5932 | 0.6218 | 0.5203 | 0.6336 | 0.5412 | 0.6258 | 0.8162 |
| Gemini-2.5-pro*(Google, 2025) | 0.5624 | - | 0.4902 | 0.5166 | 0.4300 | 0.6223 | 0.5754 | 0.6048 | 0.8482 |
| GPT-5*(OpenAI, 2025) | 0.5252 | - | 0.4741 | 0.5109 | 0.4039 | 0.6932 | 0.4550 | 0.4444 | 0.5684 |
| Claude4-opus*(Anthropic, 2025b) | 0.5139 | - | 0.3906 | 0.4513 | 0.4488 | 0.7412 | 0.4458 | 0.5744 | 0.6076 |
| GPT-O3*(OpenAI, 2024) | 0.5092 | - | 0.4379 | 0.4881 | 0.4718 | 0.6288 | 0.4302 | 0.3826 | 0.4229 |
| GPT-4V (OpenAI, 2024) | 0.4938 | 0.4915 | 0.4071 | 0.4780 | 0.4050 | 0.6308 | 0.3393 | 0.5132 | 0.8216 |
| Gemini-2.0-flash(DeepMind, 2024) | 0.4954 | 0.4801 | 0.4062 | 0.4453 | 0.3923 | 0.6112 | 0.3443 | 0.5219 | 0.8136 |
| Qwen2.5-VL-72B (Wang et al., 2024a) | 0.4904 | 0.4904 | 0.3735 | 0.4636 | 0.417 | 0.6618 | 0.3608 | 0.5272 | 0.8246 |
| InternVL2.5-78B (Chen et al., 2024f) | 0.4790 | 0.4757 | 0.3352 | 0.4981 | 0.4259 | 0.6601 | 0.3084 | 0.4800 | 0.7963 |
| GLM-4V-9B (GLM et al., 2024) | 0.4654 | 0.4474 | 0.3472 | 0.4528 | 0.3584 | 0.5596 | 0.3283 | 0.4929 | 0.7281 |
| Qwen2.5-VL-7B (Wang et al., 2024a) | 0.4243 | 0.4243 | 0.3256 | 0.4005 | 0.3547 | 0.5482 | 0.3356 | 0.4248 | 0.7474 |
| Deepseek-vl2-small(Wu et al., 2024) | 0.4142 | 0.4164 | 0.3226 | 0.4107 | 0.3627 | 0.5297 | 0.2534 | 0.4822 | 0.7192 |
| Deepseek-vl2 (Wu et al., 2024) | 0.3882 | 0.3928 | 0.3293 | 0.3383 | 0.3514 | 0.5563 | 0.2468 | 0.4309 | 0.7147 |
| Lingshu-32B (Xu et al., 2025) | 0.3541 | 0.3539 | 0.21 | 0.2723 | 0.3796 | 0.5551 | 0.30 | 0.2811 | 0.7276 |
| Qwen2-VL-2B (Wang et al., 2024a) | 0.3536 | 0.3533 | 0.2876 | 0.3319 | 0.3059 | 0.4448 | 0.2057 | 0.4171 | 0.6675 |
| Lingshu7b (Xu et al., 2025) | 0.3398 | 0.3448 | 0.2058 | 0.271 | 0.379 | 0.504 | 0.2822 | 0.2794 | 0.7078 |
| Deepseek-vl2-tiny (Wu et al., 2024) | 0.3168 | 0.3293 | 0.2660 | 0.2869 | 0.3079 | 0.4529 | 0.1817 | 0.3953 | 0.6109 |
| LLaVA-v1.5-13B (Liu et al., 2023b) | 0.2980 | 0.3008 | 0.2437 | 0.3270 | 0.2742 | 0.3177 | 0.1798 | 0.3082 | 0.4966 |
| InternVL2.5-38B (Chen et al., 2024f) | 0.3096 | 0.2994 | 0.3035 | 0.3247 | 0.2796 | 0.3109 | 0.1474 | 0.2772 | 0.4082 |
| ShareGPT4V-7B (Chen et al., 2024c) | 0.2897 | 0.2831 | 0.2232 | 0.2914 | 0.2656 | 0.4158 | 0.1476 | 0.3256 | 0.4235 |
| HuatuoGPT-vision (Chen et al., 2024a) | - | 0.279 | 0.2281 | 0.1111 | 0.2997 | 0.5649 | 0.1897 | 0.282 | 0.676 |
| LLaVA-mistral-7B (Liu et al., 2023a) | 0.2911 | 0.2731 | 0.2205 | 0.2714 | 0.2640 | 0.3740 | 0.1585 | 0.2399 | 0.4913 |
| Biomedix2(Mullappilly et al., 2024) | 0.2603 | 0.2676 | 0.171 | 0.1647 | 0.3118 | 0.4095 | 0.1563 | 0.349 | 0.5977 |
| LLaVA-v1.5-7B (Liu et al., 2023b) | 0.2648 | 0.2595 | 0.2254 | 0.2456 | 0.2288 | 0.3169 | 0.1605 | 0.3042 | 0.423 |
| InternVL2.5-4B (Chen et al., 2024f) | 0.2632 | 0.254 | 0.1895 | 0.3151 | 0.2428 | 0.2172 | 0.1336 | 0.3121 | 0.2965 |
| MedGemma-4b (Sellergren et al., 2025) | 0.195 | 0.203 | 0.1212 | 0.0915 | 0.1697 | 0.4007 | 0.1908 | 0.2339 | 0.5564 |
| SmolVLM-500M (Marafioti et al., 2025) | 0.1898 | 0.1761 | 0.1711 | 0.1602 | 0.1897 | 0.2656 | 0.0992 | 0.1417 | 0.2190 |
| SmolVLM-256M (Marafioti et al., 2025) | 0.1564 | 0.156 | 0.1397 | 0.1418 | 0.1507 | 0.2172 | 0.0912 | 0.1691 | 0.2274 |
| LLaVA-med-v1.5-7B (Li et al., 2023b) | 0.0885 | 0.0791 | 0.0372 | 0.0715 | 0.1104 | 0.1258 | 0.0466 | 0.0535 | 0.1426 |
| Only Text as Input | | | | | | | | | |
| InternVL2.5-78B (Wang et al., 2024a) | 0.3636 | 0.3839 | 0.3378 | 0.3089 | 0.3763 | 0.6606 | 0.2967 | 0.3946 | 0.8014 |
| Qwen2.5vl-72B (Wang et al., 2024a) | 0.3478 | 0.3537 | 0.2640 | 0.2784 | 0.2987 | 0.5818 | 0.3194 | 0.3016 | 0.8124 |
| InternVL2.5-4B (Chen et al., 2024f) | 0.3403 | 0.3406 | 0.2071 | 0.3023 | 0.3190 | 0.5266 | 0.2981 | 0.2645 | 0.7446 |
| GPT-4V (Achiam et al., 2023) | 0.3089 | 0.3185 | 0.2201 | 0.1687 | 0.3200 | 0.6076 | 0.2441 | 0.2844 | 0.8140 |
| Qwen2.5VL-7B(Wang et al., 2024a) | 0.3153 | 0.3097 | 0.2217 | 0.2376 | 0.2646 | 0.4900 | 0.2939 | 0.2945 | 0.7404 |
| Deepseek-vl2 (Wu et al., 2024) | 0.2981 | 0.2851 | 0.2452 | 0.1685 | 0.2916 | 0.5455 | 0.1996 | 0.3032 | 0.7227 |
| Qwen2-VL-2B (Wang et al., 2024a) | 0.2693 | 0.2814 | 0.2146 | 0.2384 | 0.2636 | 0.4195 | 0.1873 | 0.2232 | 0.6389 |
| ShareGPT4V-7B (Chen et al., 2024c) | 0.2193 | 0.2477 | 0.1940 | 0.1171 | 0.2293 | 0.3374 | 0.1439 | 0.2668 | 0.4247 |
| LLaVA-med-v1.5-7B (Li et al., 2023b) | 0.0842 | 0.0763 | 0.0349 | 0.0535 | 0.1096 | 0.1533 | 0.0398 | 0.0739 | 0.1899 |

textual information are integrated in real-world clinical diagnostic process. Each question will be assigned a label of multi-modal or text-only in the open-resource version.

## 4.2 MAIN RESULTS

2 compare the performance of 22 vision-language models on MedLesionVQA which includes 7 medical tasks aligned closely with real clinical setting, assessed through both multiple-choice and open-ended question formats. Fig 4 presents the performance of 10 representative MLLMs MedLesionVQA. In general, Gemini-2.5-pro(Google, 2025) shows the best performance across nearly all capabilities with 56.24% average accuracy. Senior physicians achieve averaged score of 73.21%, far beyond the best MLLMs. Key findings from this comprehensive comparison include:

**Insight 1: MLLMs Cannot Function as Body Surface Health Doctors.** MedLesionVQA presents significant challenges for multimodal large language models (MLLMs). The overall accuracy of representative MLLMs on our MedLesionVQA benchmark is below 57%, emphasizing the need for implementing real-world visual diagnostic tests. Although many MLLMs claim to perform at a physician's level, Tab. 2 indicates that even the best MLLM performs notably worse than primary care physicians (by 5%) and significantly worse than expert clinicians (by 17%). The primary reason of incorrect diagnosis are errors in recognizing lesion types, locations, attributes, or relationships-tasks that human doctors perform reliably while the best lesion recognition accuracy for MLLMs is only 49%. Our results from MedLesionVQA show that MLLMs frequently fail in diagnostic tasks and often struggle to align with physicians in real clinical settings. These findings underscore the need

for caution when employing MLLMs as medical practitioners and highlight the necessity to develop more advanced medical-specific MLLMs.

**Insight 2: Textual Capabilities Can Cause MLLMs to Appear More Competent Than They Are**

People often perceive MLLMs as highly knowledgeable experts and report positive experiences during question-and-answer interactions. However, our MedLesionVQA benchmark suggests that MLLMs seem more competent than they are due to their impressive text generation abilities, even when subjective questions are minimized in MedLesionVQA. A comparison between text-only and vision-text evaluations indicates that "suggestion" scores remain high regardless of the modality (82.4% vs. 81.2% with and without images). The high accuracy of treatment recommendations demonstrates that large language models can generate effective general advice, even without specialized expertise in body health images. In contrast, MLLMs perform poorly on more visually demanding tasks, such as lesion and location recognition. These findings underscore the necessity of comprehensive clinical pipeline evaluations when applying MLLMs in medical contexts.

**Insights 3: Performance Improves as Model Size Increases.** The results demonstrate a generally positive correlation between model size and performance, but with diminishing returns and notable exceptions. Models under 1B parameters (e.g., SmoMLLM-256M/500M) show limited capabilities across all tasks (scores below 0.2), while mid-scale models (1B-10B) like Qwen2-VL-2B and Deepseek-vl2-tiny (3.4B) exhibit significant performance jumps, particularly in recognition and diagnostic tasks. The GLM-4V-9B model achieves near-state-of-the-art results, rivaling much larger models with average of 0.465 compared to the 0.309 score of InternVL2.5-38B. However, scaling beyond 10B parameters shows inconsistent returns – while Qwen2-VL-72B dominates in most metrics, the InternVL2.5-78B underperforms smaller models in key areas like disease diagnosis, suggesting current architectural or training limitations

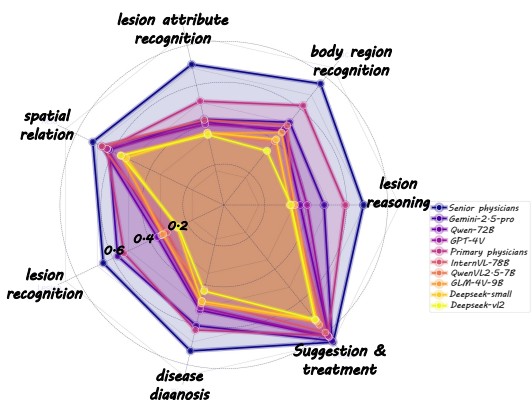

Figure 4: Results of 10 representative MLLMs across the 7 ability dimensions defined in MedLesionVQA.

in MLLMs. Generally, closed-source models consist of hundreds of billions of parameters and provide the relatively high performance.

**Insight 4: The Need to Rethink Domain-Specific Models.** These models in the medical domain mostly exhibit suboptimal performance compared to general models on a similar model-size scale. For example, the comparison between LLaVA-v1.5-7B and LLaVA-med-v1.5-7B highlights the trade-off between specialization and generalization. LLaVA-med-v1.5-7B performs 18% worse than LLaVA-v1.5-7B on the MedLesionVQA dataset, yet demonstrates superior performance on VQA-RAD. Similarly, MedGemma-4b also shows worse performance than InternVL2.5-4B. Reliance on supervised fine-tuning (SFT) for domain adaptation. Although SFT is effective, it often leads to overfitting to domain-specific patterns (e.g., radiology report generation), which may constrain a model's ability to generalize to unseen tasks—such as our body-surface–focused benchmark and even degrade underlying reasoning capabilities.

## 4.3 Per-ability Error Analysis

**Physicians v.s. Models.** We provide a general per-ability comparison between physicians and Gemini-2.5-pro, which has the best performance in MedLesionVQA. As shown in Tab. 2, several patterns emerge: (i) physicians consistently outperform Gemini-2.5-pro on visually demanding recognition and reasoning abilities (e.g., lesion and attribute recognition, region localization), especially for location recognition and disease diagnosis; (ii) the gap narrows, and in some cases reverses, on more text-centric abilities such as suggestion & treatment. We believe this quantitative breakdown already sharpens which capabilities most require architectural or training improvements.

**Error Cases.** We also analyze the error instances sampled from the GPT-4V's predictions. The distribution of these errors is illustrated in Fig 5 (a), including lack of knowledge, text misunderstanding, and judgment error, etc. The fundamental deficiency lies in the predicted answer's failure to perform targeted visual analysis. As shown in Fig 5 (b), While generically mentioning textbook features of folliculitis and acne, it critically neglected to anatomically map the documented clinical findings. This representational gap between generalized knowledge and case-specific application resulted in diagnostic inaccuracy. The specific error cases can be found in Appendix B.2 and B.3.

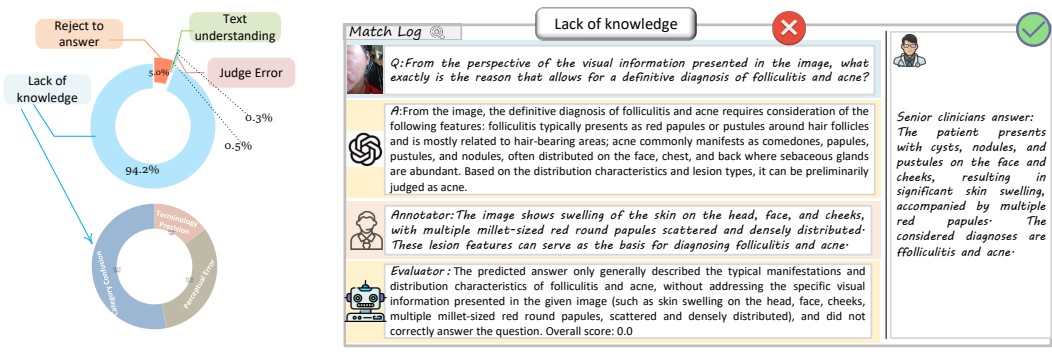

(a) Error distribution        (b) Showcase of an open-ended QA error

Figure 5: Error analysis of GPT-4V on our benchmark. Figure (a) illustrates the error analysis: the top-left panel shows the overall distribution of error cases, while the bottom-left panel displays the distribution for the 'Lack of Knowledge' category. The right figure presents an example of an incorrect response to an open-ended question.

## 5 CONCLUSION

In this paper, we propose MedLesionVQA, a large-scale and body surface oriented benchmark evaluating the lesion, region, diagnosis, and treatment-related recognition and reasoning ability for medical MLLMs. MedLesionVQA contains 12K body lesion images with expert-level fine-grained annotations of 96 prevalent dermatological diseases, 94 distinct lesion types and 110 body regions. The evaluation dimension of MedLesionVQA is built on basis of 7 multimodal stepwise visual diagnostic abilities, including lesion recognition, lesion attribute recognition, body region recognition, lesion spatial relation recognition, lesion reasoning, disease diagnosis and suggestion & treatment, which ensure the alignment with the authentic clinic senary. Mainstream MLLMs are evaluated on the benchmarks, and Gemini-2.5-pro has the best score of 56.24. Furthermore, senior and primary physicians are invited to answer the questions of benchmark and obtain score of 61.44 and 73.21, respectively. The results show that there is large improvement for MLLMs on the benchmark and indicates significant challenges and medical specialization of the MedLesionVQA.

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

# MedLesionVQA: A Multimodal Benchmark Emulating Clinical Visual Diagnosis for Body Surface Health

# Supplementary Materials

## A  DATASET DETAILS

### A.1  DETAILS FOR DATA DISTRIBUTION.

All annotation data are reviewed by another group of expert physicians, and the qualified standard is that the recall and precision of the sampled data in each annotation dimension reach 95%, respectively. Among all images obtained through annotation, taking into account the coverage of body region, lesion, and disease, we screen out a total of 12K images including 10K images with abnormalities and 2K images without abnormalities. The histogram distribution of lesion, disease and body region is shown in Fig. 6, which is used to illustrate the annotation information density. Taking the left sub-figure as example, the horizontal axis represents the number of images containing a certain lesion, while the vertical axis represents the number of these type of lesions, which indicates that most type of lesions has at least 50 images and there is enough lesion annotations and there is less issue of long-tail distribution.

Fig. 7 offers a detailed view of the data distribution for different clinical-oriented abilities. Taking the top-left image as an example: This multi-layer ring chart illustrates the distribution of recognition questions in the test dataset across four main categories. The outer ring shows the total number of questions for each category: Region Recognition contains 3,986 questions, Attribute Recognition has 3,508 questions, Lesion Recognition includes 3,340 questions, and Spatial Relation comprises 1,133 questions. The inner ring further breaks down each category into different types, including multiple-choice, judgment, fill-in-the-blank, and short-answer questions, highlighting their relative proportions within each category.

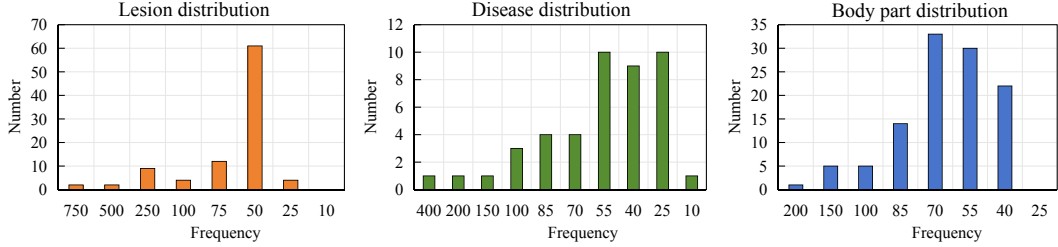

Figure 6:  Histogram of lesion, disease, and body region distribution of MedLesionVQA. The left figure shows that most lesions has at least 50 images.

### A.2  DETAILS FOR LEXICAL TREES IN ANNOTATION PROTOCOL

We have respectively construct the corresponding lexical trees for part division. The detailed information of the lexical trees are listed below, in Tab. 3 and 4. Visible body parts are annotated with the secondary node in the level-2 list , while abnormal body parts are annotated with more refined leaf node in the level-2, level-3 or level-4 list according to the minimum body division. For example, the leaf node of upper eyelid is in the path (head -> face -> periorbital area -> upper eyelid) at the fourth level. The leaf node of anterior neck is in the path (neck -> anterior neck) at the second level.

Besides the lexicon of body regions, we construct the lexicon list of lesion, disease and lesion attributes, as shown in Tab. 5, 6 and 7

Table 3: Lexical tree for body regions.

| Level 1 | Level 2 | Level 3 | Level 4 |
|---------|---------|---------|---------|

| | | | |
|---|---|---|---|
| head | scalp | top | |
| | | temporal region | |
| | | occipital region | |
| | forehead | | |
| | face | cheek | |
| | | perioral area | |
| | | chin | |
| | | periorbital area | upper eyelid |
| | | | lower eyelid |
| | | | supraorbital arch |
| | | perinasal area | |
| | | nose | |
| | ear | | |
| | postauricular area | | |
| neck | anterior neck | | |
| | lateral neck | | |
| | posterior neck | | |
| trunk | chest | breast | |
| | | submammary area | |
| | | nipple | |
| | | areola | |
| | back | | |
| | waist | | |
| | abdomen | periumbilical area | |
| | | groin | |
| | lateral trunk | axilla | |
| | | iliac region | |
| | shoulder | | |
| upper limb | upper arm | flexor side | |
| | | extensor side | |
| | elbow | extensor side | |
| | | cubital fossa | |
| | forearm | flexor side | |
| | | extensor side | |
| | hand | wrist | |
| | | dorsum of hand | |
| | | palm | |
| | | interdigital space | |
| | | thenar space | |
| | | finger | lateral side of finger |
| | | | palmar side of finger |
| | | | dorsal side of finger |
| | | | tip of finger |
| lower limb | buttock | | |
| | thigh | medial side | |
| | | lateral side | |
| | knee | | |
| | popliteal fossa | | |
| | leg | extensor side | |
| | | flexor side | |
| | foot | dorsum of foot | |
| | | sole | heel |
| | | | plantar aspect of foot |
| | | | forefoot |
| | | first metatarsophalangeal joint area | |
| | | lateral border of foot | |

| | | ankle | |
|---|---|---|---|
| | | interdigital space of toes | |
| | | toe | dorsal side of toe |
| | | | plantar side of toe |
| | | | lateral side of toe |
| | | | tip of toe |
| nail | nail plate | | |
| | nail bed | | |
| | nail root | | |
| | nail fold | | |
| | nail groove | | |
| perineum | anus | | |
| | perianal skin | | |
| | female external genitalia | labia majora | |
| | | labia minora | |
| | | clitoris | |
| | | vaginal orifice | |
| | male external genitalia | penis | glans penis |
| | | | external urethral orifice in male |
| | | | coronal sulcus |
| | | | prepuce |
| | | | penile shaft |
| | | scrotum | |
| skin of undetermined location | upper limb or lower limb | | |
| mucosa of undetermined location | | | |

## A.3 Prompts for Automatic Constructing QAs

Table 8,9 systematically outline the task framework for automatically constructing question-answering (QAs). The QA construction prompts for lesion recognition, location recognition, lesion reasoning, spatial relation, disease diagnosis and suggestion & treatment are shown in Table 8. Lesion attributes contain size, color, shape, distribution, quantity, and boundary. We design QA construction prompts for each attribute, as shown in Table 9.

## A.4 Explanation of 7 Clinic-oriented Mulitmodal Abilities

To further illustrate the evaluation dimension of MedLesionVQA, we give detailed explanation for each evaluated ability, as follows:

**Lesion Recognition**. Skin lesion is an abnormal condition on the surface of body skin. For example, macule lesion represents a change in surface color without elevation or depression, while papule lesion is a circumscribed and solid elevation of skin, varying in size. Thus, lesion recognition requires MLLMs the perception ability of visual semantic information of disease images.

**Attribute Recognition**. Lesion attributes include its color, size, quantity, boundary clarity, spatial distribution, and geometric shape. There is no need to describe all attributes but some key attributes for each lesion. The key attributes of bulla lesion include its size, shape, quantity, and spatial distribution, and the reason for ignoring its color attribute is that nearly all bulla has a typical color of transparent skin tones. Attribute recognition demands the perception ability of detailed visual information and understanding of general world knowledge.

**Location Recognition**. Region represents human body regions, such as head, face, ear, hand, foot, etc. We expect medical MLLMs to recognize body regions like clinical doctors do. Moreover, body region information is related to disease diagnosis because some diseases may frequently occur in certain body regions.

Table 4: Lexical tree for oral cavity.

| Level 1 | Level 2 | Level 3 | Level 4 |
|---|---|---|---|
| oral mucosa | labial mucosa | upper labial mucosa | upper labial frenum |
| | | lower labial mucosa | lower labial frenum |
| | buccal mucosa | orifice of parotid duct | |
| | | occlusal line of teeth | |
| | | buccal frenum | |
| | mucosa of retromolar area | | |
| tongue | tip of tongue | lingual papillae on the tip of tongue | |
| | dorsum of tongue | lingual papillae on the dorsum of tongue | |
| | root of tongue | lingual papillae on the root of tongue | |
| | lateral border of tongue | lingual papillae on the lateral border of tongue | |
| | ventral surface of tongue | sublingual veins | |
| | sublingual region | lingual frenum | |
| | | sublingual caruncle | |
| | | sublingual fold | |
| palate | hard palate | | |
| | soft palate | | |
| pharynx | posterior pharyngeal wall | | |
| | uvula | | |
| | palatoglossal arch | | |
| | palatopharyngeal arch | | |
| tonsil/adenoid | tonsil | | |
| | adenoid | | |
| lip | upper lip | | |
| | lower lip | | |
| | margin of upper lip | | |
| | margin of lower lip | | |
| | angle of mouth | | |
| ingiva | gingiva of central incisor | | |
| | gingiva of lateral incisor | | |
| | gingiva of canine | | |
| | gingiva of premolar | | |
| | gingiva of molar | | |
| teeth | central incisor | | |
| | lateral incisor | | |
| | canine | | |
| | premolar | | |
| | molar | | |

Table 5: Lesion list.

| Lesions |
| --- |
| macule, patch, papule, plaque, mass, vesicle, bulla, pustule, wheal, alopecia, nevus, scale, scale, crust, fissure, scar, pigmentation, depigmentation, swelling, erosion, ulcer, hypertrophy, breakage, peeling, hypopigmentation, blood blister, excrescence, keratotic papule, keratotic plaque, erythema, striae atrophicae, comedone, maculopapule, fissure, rupture, maceration, excoriation, exudation, dryness, lichenification, thickening of the skin, topical preparation, papule/macule, papule/vesicle, macule/vesicle, patch/plaque, unidentifiable lesion (poor image quality), unidentifiable lesion (difficult to classify), unidentifiable lesion (possible physiological nature), opening, reddening, keratin plug, blackhead, whitehead, bleeding, purulent discharge, elevated edge, xerosis capillorum, trichoptilosis, pili annulati, canities, thickening, atrophy, roughness, onycholysis, absence, longitudinal fissure, longitudinal stripe, melanonychia striata, transverse stripe, punctate depression, unevenness, subungual hemorrhage, color change of the nail plate, alopecia patch, hypotrichosis, absence of hair, receding hairline, loss of eyebrows, sparse eyebrows, loss of eyelashes, nodule, ecchymosis, petechia, striae, pseudomembrane, frenum rupture, recession, groove and fissure, desquamation of tongue coating, thickening of tongue coating, tooth mark, tonsillar hypertrophy, adenoidal hypertrophy |

Table 6: Disease list.

| Diseases |
| --- |
| acne, rosacea, lupus miliaris disseminatus faciei, seborrheic dermatitis, scalp psoriasis, psoriasis, pityriasis rosea, eczema, tinea corporis, folliculitis, androgenetic alopecia, telogen effluvium, alopecia areata, pseudopelade, trichotillomania, tinea capitis, systemic lupus erythematosus, syphilitic alopecia, white hair, contact dermatitis, lichen simplex chronicus, tinea manus, tinea pedis, acute eczema, chronic eczema, asteatotic eczema, nummular eczema, pompholyx, stasis dermatitis, auto-sensitive dermatitis, progressive pigmented purpuric dermatosis, atopic dermatitis, scabies, exfoliative keratolysis, palmoplantar pustulosis, palmoplantar keratoderma, onychomycosis, onychodystrophy, tinea cruris, malassezia folliculitis, urticaria, urticarial vasculitis, dermatographism, cold contact urticaria, chronic urticaria, chronic spontaneous urticaria, cholinergic urticaria, nevus, skin tag, herpes zoster, herpes simplex, impetigo, varicella, papular urticaria, prurigo, verruca plantaris, corn, callus, verruca vulgaris, verruca filiformis, verruca plana, pruritus cutis, pediculosis, insect bite dermatitis, keratosis pilaris, lichen spinulosus, pityriasis rubra pilaris, post-inflammatory hypopigmentation, pityriasis alba, pityriasis versicolor, nevus anemicus, achromic nevus, vitiligo, alopecia, hyperhidrosis, ichthyosis, scleroderma, molluscum contagiosum, diaper dermatitis, pemphigus, bullous pemphigoi, melasma, freckles, cutaneous candidiasis, furuncle, carbuncle, paronychia, erysipelas, cellulitis, dermatophytosis, lichen planus, basal cell carcinoma, squamous cell carcinoma, malignant melanoma, keloid scar, hypertrophic scar |

Table 7: Value lists of attributes

| Attributes | Value List |
| --- | --- |
| Size | needlepoint-sized, sesame-seed-sized, millet-grain-sized, rice-grain-sized, mung-bean-sized, soybean-sized, fingernail-sized, coin-sized, walnut-sized, palm-sized, x cm in diameter, y cm * z cm |
| Color | white, pinkish white, red, light red, pink, dark red, yellow, purple, dark purple, gray, black, brown, flesh-colored |
| Shape | subcircular, subelliptical, subspherical, subhemispherical, elongated strip-shaped, irregular, annular, cauliflower-like, spider-like, target-shaped, crab claw-like |
| Quantity | 1, 2, 3, a few, multiple |
| Distribution | scattered, dense, cluster-like, symmetrical, zonal, reticular, fused, partially fused, fused into a sheet, adjacent, diffuse, localized |
| Boundary | well-defined, poorly-defined, relatively well-defined |

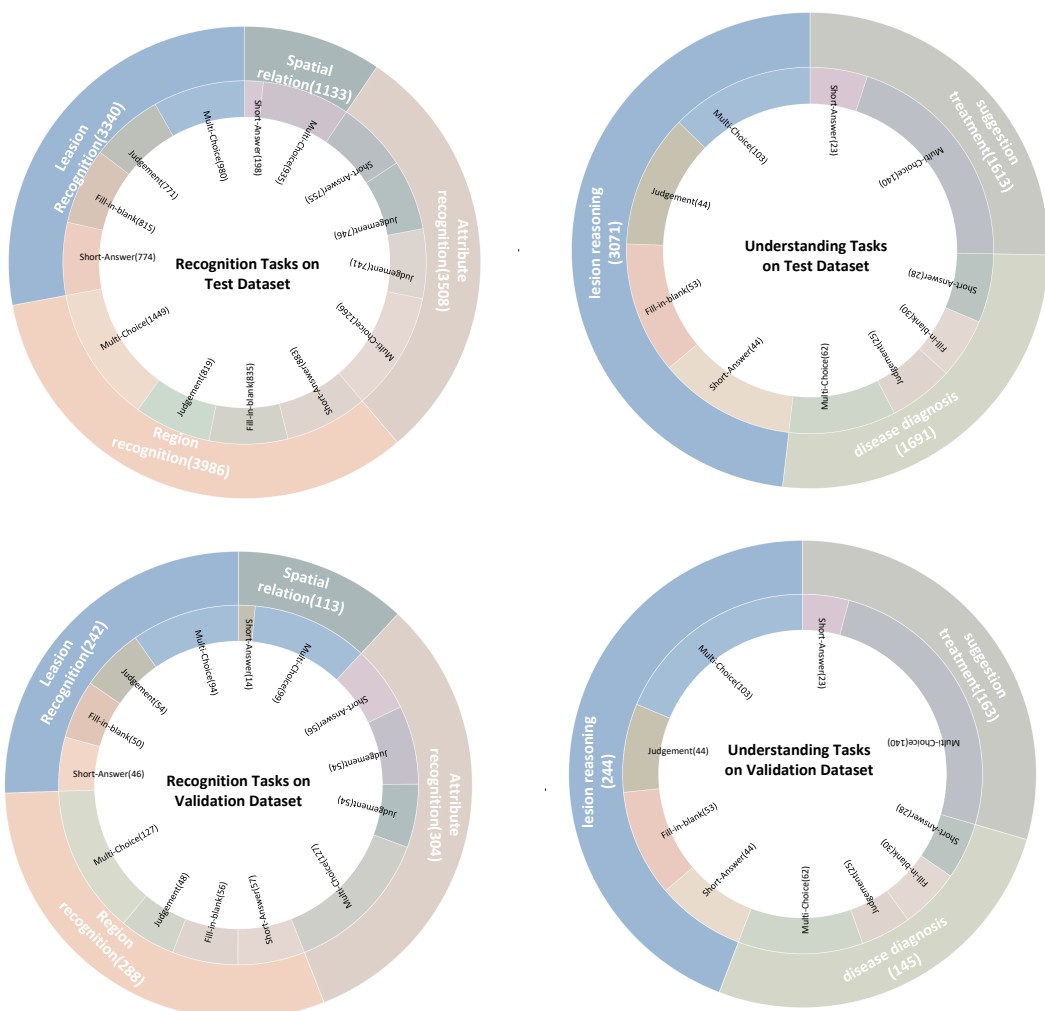

Figure 7: Data distribution of MedLesionVQA benchmark.

**Spatial Relation**. Spatial relationship between lesions serves as evaluation dimension when there are more than one lesions on affected skin area and can be a hint for analysis of primary and secondary lesions. Spatial relation evaluates the complex visual understanding about different lesion entities.

**Lesion Reasoning**. Lesion reasoning evaluates MLLMs' capacity to deduce pathological mechanisms from visual information. Furthermore, it also involves the analysis of how primary lesions evolve (e.g., papules progressing to pustules in folliculitis) or how secondary lesion changes (scaling, scabs) are introduced.

**Disease Diagnosis**. Disease diagnosis requires MLLMs to exploit both multi-dimensional visual information, including lesion, its attributes, body region, and spatial relationships, and cross-referencing medical knowledge to map clinical manifestations to potential pathologies. Thus, disease diagnosis evaluates the comprehensive visual recognition and reasoning ability of MLLMs.

**Suggestion & Treatment**. Suggestion and treatment recommendations should align with evidence-based medical knowledge while incorporating patient-personalized context, including lesion severity, attributes, and region, which is reflected in lesion images. Thus, this evaluation dimension requires comprehensive multimodal reasoning ability integrating knowledge and visual information.

Table 8: Automatic QA constructing prompts

| Ability Category | Prompt Description |
|---|---|
| Lesion Recognition | Task Description: You will be provided with a piece of medical information related to a picture. Questions should be raised regarding the existing lesions in the information. Assume that all the information is sourced from the picture, and the recognition ability of the model for the lesions shown in the picture needs to be examined. The model is required to answer with the lesion terms as the answers.
Medical Information: { }
Lesions: { } |
| Lesion Reasoning | Task Description: You will be provided with a piece of medical information related to a picture. Please pose questions regarding the reasoning process for lesion identification, assuming all information is derived from the image. The questions should assess the model's ability to recognize lesions based on attribute information and other key information obtained from the image in the reasoning process
Medical Information: { }
Lesions: { } |
| Spatial Relation | Task Description: You will be provided with a piece of medical information related to a picture. Please pose questions regarding the spatial relationships between lesions, assessing the model's ability to understand the spatial relationships among multiple lesions in the image. The model should respond with related terms as the answer.
Medical Information: { }
Spatial relationship: { } |
| Disease Diagnosis | Task Description: You will be provided with a piece of medical information related to a picture. Provide you with a piece of medical information for a picture. Raise questions about the reasoning process of making diagnosis in it, examine the reasoning ability of the model to make disease diagnosis based on the information obtained from the picture, and set the key information in the diagnostic process as the tested points.
Medical Information: { }
Diagnostic reasoning: { } |
| Location Recognition | Task Description: You will be provided with a piece of medical information related to a picture. Please pose questions regarding the location (body region) where the lesion appears, assuming all information is derived from the image. The questions should assess the model's ability to recognize the location of the lesion area in the image, and the model needs to respond with body regions as the answer.
Medical Information: { }
Regions: { } |
| Suggestion & Treatment | Task Description: You will be provided with a piece of medical information related to a picture. Please pose questions regarding the content of suggestion & treatment, assessing the model's medical knowledge ability to make such suggestions based on information obtained from the image.
Medical Information: { }
Suggestions: { } |

## A.5 JUDGE-LLM, PROMPT DESIGN, AND RELIABILITY ANALYSIS

### A.5.1 SCORING PROTOCOL AND PROMPT

For the open-ended image–QA evaluation, we use GPT-4 as a judge-LLM to score model responses. Given a question, its reference (gold) answer, and a model-predicted answer, the judge-LLM assigns a score from $\{0.0, 0.5, 1.0\}$, corresponding to incorrect, partially correct, and fully correct, respectively. The judge is instructed to (1) compare the predicted answer with the reference answer under the context of the question, (2) provide a short explanation of the decision, and (3) output the final score in a machine-readable form as `<result>score</result>`.

Table 9: Automatic QA constructing prompts for lesion attribute

| Subtask | Prompt Description |
|---|---|
| Lesion Size | Task Description: You are provided with a paragraph of medical exam point information and supplementary medical information about an image. Please pose questions regarding the size of the lesion, assessing the model's ability to identify the size attribute of the lesion in the image. The model should respond with size terms as the answer. Add constraints in the question stem to specify whether the answer should use analogy (e.g., "pinpoint-sized", "grain-sized") or specific measurements (in mm or cm). Medical Exam Point Information: Attribute - Size: {} |
| Lesion Color | Task Description: You are provided with a paragraph of medical exam point information and supplementary medical information about an image. Please pose questions regarding the color of the lesion, assessing the model's ability to identify the color attribute of the lesion in the image. The model should respond with color terms as the answer. If the lesion term includes color information, do not reveal the color details in the question. Medical Exam Point Information: Attribute - Color: {} |
| Lesion Shape | Task Description: You are provided with a paragraph of medical exam point information and supplementary medical information about an image. Please pose questions regarding the shape of the lesion, assessing the model's ability to identify the shape attribute of the lesion in the image. The model should respond with shape terms as the answer. Medical Exam Point Information: Attribute - Shape: {} |
| Lesion Distribution | Task Description: You are provided with a paragraph of medical exam point information and supplementary medical information about an image. Please pose questions regarding the distribution of the lesion, assessing the model's ability to identify the distribution attribute of the lesion in the image. The model should respond with distribution terms as the answer. Medical Exam Point Information: Attribute - Distribution: {} |
| Lesion Quantity | Task Description: You are provided with a paragraph of medical exam point information and supplementary medical information about an image. Please pose questions regarding the quantity of the lesion, assessing the model's ability to identify the quantity attribute of the lesion in the image. The model should respond with quantity terms as the answer. If the description of the quantity is vague, ignore the question type requirements and only construct multiple-choice questions to enable the model to answer more definitively. Medical Exam Point Information: Attribute - Quantity: {} |
| Lesion Boundary | Task Description: You are provided with a paragraph of medical exam point information and supplementary medical information about an image. Please pose questions regarding the boundary of the lesion, assessing the model's ability to identify the boundary attribute of the lesion in the image. The model should respond with boundary terms as the answer. Medical Exam Point Information: Attribute - Boundary: {} |

The scoring is purely text-based: the judge-LLM only observes the question text, the reference answer, and the model's predicted answer, but not the image itself. Thus, the evaluation probes the semantic correctness of the generated textual answers, and does not depend on any visual capability of the judge-LLM.

### A.5.2 RUBRIC AND DISAMBIGUATION RULES

To reduce ambiguity and avoid overly strict grading on surface-form mismatches, we provide the judge-LLM with a detailed scoring rubric. The final prompt used in all reported experiments contains the following guidelines in Table 10):

Table 10: English translation of the judge-LLM prompt used for open-ended scoring.

| Section | Content |
|---|---|
| Overall instruction | Based on the question, the correct answer, and the predicted answer, evaluate the correctness score of the predicted answer. |
| Evaluation method | • Decide whether the predicted answer correctly answers the question, provide scoring details, and finally give an overall score.
• The supplementary scoring rules below specify the scoring standards for different types of questions, and are used to handle situations where the expressions in the predicted answer and the reference answer are not exactly the same.
• The score must be one of the following values: 0.0 (completely incorrect), 0.5, or 1.0 (completely correct).
• Output the final score using the format `<result>score</result>`, for example: `<result>0.5</result>`. |
| Question and answers | Question: `{question}`
Reference answer: `{reference_answer}`
Predicted answer: `{predicted_answer}` |
| Supplementary scoring rules: Color | Colors are unified by color families, with explicit scoring rules. Same color: 1.0; same color family: 0.5; different color families: 0.
Red family: red-related colors such as red, light red, dark red, deep red, etc.; purple-related colors such as purple, dark purple, deep purple, purplish red, etc.
Dark family: brown-related colors such as brown, dark brown, dark tan, light tan, etc., and black-related colors such as black. |
| Supplementary scoring rules: Shape | Shapes are categorized, with explicit scoring rules. Exactly the same: 1.0; same category: 0.5; different categories: 0.
Circular category: round-like, ellipse-like.
Rectangular category: rectangle, square, quadrangular.
Irregular category: bran-like (furfuraceous) shapes. |
| Supplementary scoring rules: Size | If comparative descriptions or concrete numeric values correspond to a common-sense similar size, give 1.0; if they are close, give 0.5; if the gap is too large, give 0. |
| Supplementary scoring rules: Lesions | "Blood scab" and "crust" are treated as equivalent to "scab"; "scale" is treated as equivalent to "desquamation". |
| Supplementary scoring rules: Diseases | If the answer is a synonym of the same disease, it should be regarded as a correct answer. |

- **Color.** Colors are grouped into coarse color families, and scores are assigned as: (i) exactly the same color: 1.0; (ii) same color family but not exactly the same term: 0.5; (iii) different color family: 0.0. For example, "light red" and "dark red" are in the red family, and "purple" and "dark purple" are in the purple family; browns (e.g., "brown", "dark brown") and blacks are grouped into a dark-color family.

- **Shape.** Shapes are grouped into coarse categories, with: (i) exactly the same shape: 1.0; (ii) same category (e.g., different but related shapes): 0.5; (iii) different categories: 0.0. For instance, circle-like and ellipse-like descriptions are in one category, rectangle- and square-like descriptions are in another, and irregular shapes (e.g., "bran-like / furfuraceous") are treated separately.

- **Size.** For size descriptions, common-sense consistency is used: (i) fully consistent or equivalent units: 1.0; (ii) close but not identical size: 0.5; (iii) substantially different scale: 0.0.

- **Lesions / Morphology.** Certain clinically equivalent terms are treated as the same category, e.g., "blood scab" and "crust" are both mapped to "scab", and "scale" is treated as equivalent to "desquamation".

- **Diseases / Diagnoses.** If the predicted answer uses a standard synonym or an alternative yet clinically equivalent name for the same disease as in the reference answer, the response is considered fully correct (1.0).

These rules explicitly address the disagreement modes that we observed in a preliminary analysis, where the judge-LLM tended to be overly strict on attributes such as color and size (e.g., treating "pink" vs. "skin tone", or "pinpoint" vs. "millimeter" as completely different). After incorporating the

above rubric into the prompt, we fix this prompt and use it unchanged for all experiments reported in the main paper and appendix.

### A.5.3 Agreement with Human Expert

To quantify the reliability of GPT-4 as a judge-LLM, we evaluate its agreement with a senior physician on all open-ended questions in the test set (18,344 image–QA pairs). Both the human expert and the judge-LLM assign scores from $\{0.0, 0.5, 1.0\}$ using the same rubric.

**Exact-match consistency and average score difference.** Let $s_{\text{human}} \in \{0.0, 0.5, 1.0\}$ and $s_{\text{LLM}} \in \{0.0, 0.5, 1.0\}$ be the scores for a given sample, and define the absolute score difference

$$\Delta s = |s_{\text{human}} - s_{\text{LLM}}| \in \{0, 0.5, 1.0\}.$$

On the 18,344 open-ended QA pairs, the distribution of $\Delta s$ is:

| $\Delta s$ | 0 | 0.5 | 1.0 |
|---|---|---|---|
| Proportion | 91.43% | 6.45% | 2.12% |

The average absolute score difference is therefore

$$\overline{\Delta s} = 0 \times 0.9143 + 0.5 \times 0.0645 + 1.0 \times 0.0212 = 0.053.$$

We define the consistency rate as $1 - \overline{\Delta s}$, which yields a final consistency of

$$1 - \overline{\Delta s} = 0.947 \approx 95\%.$$

In other words, GPT-4 exactly matches the human score on over 91% of the samples, and differs by at most one tier (0.5) on the vast majority of the remaining cases.

**Quadratic-weighted Cohen's Kappa.** To further assess inter-rater reliability, we compute the quadratic-weighted Cohen's Kappa $\kappa_w$ between the human expert and GPT-4 on the same set of open-ended questions. Using the standard quadratic weighting scheme for ordinal categories, we obtain

$$\kappa_w = 0.8825,$$

which is typically interpreted as high or "almost perfect" agreement between the judge-LLM and the human expert.

**Empirical reliability.** The high exact-match rate (95%), the very small average absolute score difference ($\overline{\Delta s} = 0.053$), and the quadratic-weighted Cohen's Kappa of $\kappa_w = 0.8825$ demonstrate that GPT-4's scores closely track those of a senior physician. This indicates that, despite being a proprietary model, the judge-LLM behaves as a stable and reliable proxy for expert assessment in our setting.

## B  More Results

In this section, we present the comprehensive experimental results, followed by a detailed case study analyzing representative examples of model outputs.

### B.1  Evaluation of Few-shot and CoT Prompting

Our primary evaluation is conducted in a "zero-shot" setting, which aligns with the standard practice of prominent benchmarks such as MMBench (Xu et al., 2023), OmniMedVQA (Hu et al., 2024), and MMMU (Yue et al., 2024). This approach effectively assesses the model's intrinsic, out-of-the-box capabilities. To further explore the upper bounds of the model's performance and its adaptability, we also conducted supplementary experiments, incorporating few-shot and CoT prompting. The results are presented in Tab. 11. The following results show that there is an increasing trend but no significant differences.

Table 11: Evaluation of Few-shot and CoT Prompting

| Model | AVG_val | Lesion | Loc. | Attr. | Rela. | Diag. | Sugg. |
|---|---|---|---|---|---|---|---|
| **Senior physicians** | **0.7321** | **0.6826** | **0.7583** | **0.7046** | **0.7102** | **0.7313** | **0.8574** |
| **Primary physicians** | **0.6144** | **0.5932** | **0.6218** | **0.5203** | **0.6336** | **0.6258** | **0.8162** |
| *GPT-4.1* | | | | | | | |
| (Zero-shot) | 0.5276 | 0.4487 | 0.5303 | 0.4308 | 0.6401 | 0.4879 | 0.8304 |
| (1-shot) | 0.5526 | 0.4791 | 0.4623 | 0.4700 | 0.6525 | 0.6064 | 0.8634 |
| (3-shot) | 0.5582 | 0.4808 | 0.5338 | 0.4614 | 0.6260 | 0.5477 | 0.8615 |
| (5-shot) | 0.5536 | 0.4332 | 0.5404 | 0.4682 | 0.6566 | 0.5666 | 0.8565 |
| (CoT) | 0.5377 | 0.4235 | 0.5706 | 0.4221 | 0.6348 | 0.4868 | 0.8245 |
| *Gemini-2.5 Pro* | | | | | | | |
| (Zero-shot) | 0.5479 | 0.4435 | 0.5052 | 0.4269 | 0.6516 | 0.6368 | 0.8716 |
| (1-shot) | 0.5506 | 0.5018 | 0.4660 | 0.4540 | 0.6890 | 0.5856 | 0.8694 |
| (3-shot) | 0.5550 | 0.4774 | 0.5138 | 0.4646 | 0.6017 | 0.5661 | 0.8398 |
| (5-shot) | 0.5588 | 0.4775 | 0.5256 | 0.4840 | 0.6404 | 0.4415 | 0.5573 |
| (CoT) | 0.5441 | 0.4469 | 0.5360 | 0.4195 | 0.6254 | 0.6186 | 0.8378 |

## B.2 ERROR ANALYSIS

For a comprehensive understanding of the model's operational strengths and weaknesses, we meticulously examined error instances sampled from GPT-4V's predictions. The distribution of these errors is illustrated in Fig 5 (a). The fundamental deficiency lies in the predicted answer's failure to perform targeted visual analysis. As shown in Fig 5 (b), While generically mentioning textbook features of folliculitis and acne, it critically neglected to anatomically map the documented clinical findings. This representational gap between generalized knowledge and case-specific application resulted in diagnostic inaccuracy. In addition, Current models lack explicit medical concept grounding, meaning visual features (e.g., "millet-sized papules") are not rigorously verified against diagnostic criteria (e.g., folliculitis vs. acne). If the model has low confidence in its answer, it may either refuse to respond or provide an irrelevant reply.

## B.3 ERROR CASES

**Judge Error.** As shown in Fig. 12, the judgment model incorrectly identifies a correct answer as erroneous. In other words, the judge model mistakenly flags a valid response as erroneous. This type of error arises during the assessment phase rather than during the answer generation phase, and then potentially affects the overall accuracy of the evaluation by unfairly penalizing correct answers.

**Text Misunderstanding.** As shown in Fig. 11, GPT-4V fails to correctly understand the question and generate erroneous answers. This indicates that the model struggles with accurately interpreting the semantic meaning or the specific intent behind the input question. This misunderstanding can lead to responses that are irrelevant, partially correct, or completely incorrect, highlighting limitations in the natural language comprehension capabilities of the model in this context.

**Reject to answer.** As shown in Fig. 8, GPT-4V outputs irrelevant responses or declines to answer certain questions. This behavior suggests that the model may encounter difficulties in comprehending complex or ambiguous queries, leading it either to produce answers unrelated to the question or to withhold a response altogether.

**Lack of knowledge.** We further categorize Lack of Knowledge into three error types (as shown in Fig. 5)(a): Category Confusion, Perceptual Error, and Terminology Precision. **1) Category Confusion**: The model fails to correctly distinguish between categories or hierarchies of concepts, leading to misclassification or illogical associations (Fig. 9). **2) Perceptual Error**: The model struggles with spatial, physical, or contextual reasoning, resulting in implausible descriptions or interpretations of real-world scenarios (Fig. 10). **3) Terminology Precision.** The model uses imprecise, overly broad, or incorrect terms, especially in technical domains (Fig. 13).

## C  LIMITATIONS AND POTENTIAL RISKS.

Our proposed MedLesionVQA is the first clinically-oriented benchmark designed to assess whether current state-of-the-art MLLMs can function similarly to real physicians and to guide the development of future MLLMs. However, publicly releasing the dataset may lead to leakage, potentially compromising its future effectiveness as a gold-standard benchmark.

**Error: Reject to Answer** 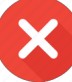

*Question:*
*Based on the visual information presented in the images, why can a definitive diagnosis of fungal skin disease (Malassezia folliculitis) be made?*

*Ground Truth Answer:*
*✓ The images show multiple pinpoint-to-rice-grain-sized, flesh-colored to pink, roundish papules on the neck and chest, with secondary erythematous manifestations. These characteristics are consistent with the clinical presentation of Malassezia folliculitis, thus confirming the diagnosis.*
*Model Prediction:*
*✗ Visual images alone are insufficient for diagnosis. Requires clinical history and lab tests.*

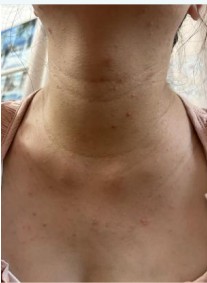

**Scoring Breakdown:**
*Match Log:*
*✗ Fundamental Misalignment:*

*Expected: Justification of visual diagnosis*

*Provided: Refutation of visual diagnosis*

**Composite Score:**
*<result>0.0</result>*

Figure 8: An example of reject to answer.

**Error: Category Confusion** 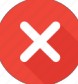

Question:
The diseases diagnosed in the image include eczema and ______.

Ground Truth Answer:
  ✓ contact dermatitis
Model Prediction:
  ✗ skin infection

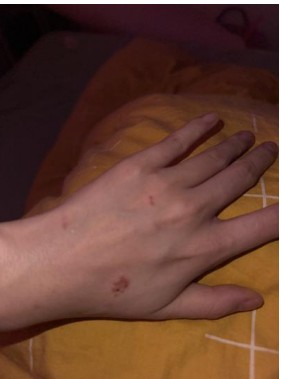

**Scoring Breakdown:**
Match Log:
Judgment: The predicted answer is incorrect. The question specifically requested identification of another diagnosis besides eczema evident in the image, with "contact dermatitis" being the correct answer. "Skin infection" does not fulfill the required diagnostic precision.

**Composite Score:**
<result>0.0</result>

Figure 9:  An example of category confusion.

**Error: Perceptual Error** 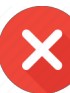

*Question:*
*The key lesion confirming melanocytic nevus in the image is ____"*

*Ground Truth Answer:*
*✓ A rice-grain-sized black nevus*
*Model Prediction:*
*✗ Pigmentation*

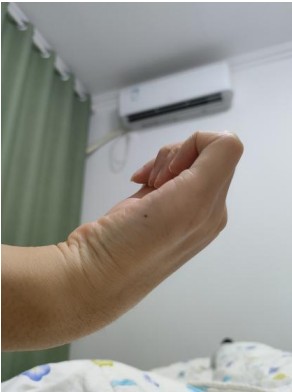

**Scoring Breakdown:**
*Match Log:*
*Judge: The predicted answer only generally mentioned pigment deposition and did not explicitly point out the specific key lesion as in the correct answer, which states "a millet-sized black nevus," thus failing to correctly answer the question.*
*Scoring details: The prediction did not provide an accurate and specific description of the "key lesion for diagnosing pigmented nevus in the image" but instead gave a broad concept, so the score is 0.0.*

**Composite Score:**
*<result>0.0</result>*

Figure 10: An example of perceptual error.

## Error: Text Misunderstanding 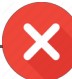

*Question:*
*From the perspective of the visual symptom information presented in the image, what exactly allows for a definitive diagnosis of fungal skin disease—tinea cruris?*

*Ground Truth Answer:*
*✓ The diagnosis is based on lesions including flat macules and raised papules, specifically pigmented macules. The macules are numerous, dark red in color, and round, near-oval, or irregular in shape, with a scattered distribution.*
*Model Prediction:*
*✗ From the perspective of the visual symptom information presented in the image, the diagnosis of skin diseases requires consideration of the lesion's morphology, distribution, color, presence of exudate or crusting, as well as the patient's medical history and symptoms.*

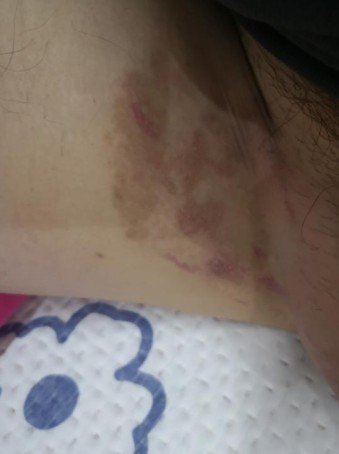

**Scoring Breakdown:**
*Match Log:*
*Judge: The predicted answer only generally mentioned pigment deposition and did not explicitly point out the specific key lesion as in the correct answer, which states "a millet-sized black nevus," thus failing to correctly answer the question.*
*Scoring details: The prediction did not provide an accurate and specific description of the "key lesion for diagnosing pigmented nevus in the image" but instead gave a broad concept, so the score is 0.0.*

**Composite Score:**
*<result>0.0</result>*

Figure 11: An example of text misunderstanding error.

**Error: Judge Error**

Question:
The size of the alopecia areata patch in the image is ___ cm × ___ cm.

Ground Truth Answer:
 ✓ 5 * 1.5
Model Prediction:
 ✗ 4 * 2

**Scoring Breakdown:**
Match Log:
The predicted values for both dimensions deviate from the correct measurements.

No partial credit is awarded as the answer is entirely incorrect.

**Composite Score:**
<result>0.0</result>

Comment:
Predicted dimensions (4 × 2 cm) are within <20% error margin of correct measurements (5 × 1.5 cm):

**Length: 4 vs. 5 cm (20% under)**

**Width: 2 vs. 1.5 cm (clinically negligible in alopecia areata assessment).**

Figure 12: An example of text understanding error.

Figure 13: An example of terminology precision error.

