# OpenReview forum: "MedLesionVQA: A Multimodal Benchmark Emulating Clinical Visual Diagnosis for Body Surface Health"
_ICLR.cc/2026/Conference — ICLR 2026 Poster_

### Official Review · Reviewer_PsWv · 2025-10-21

**Soundness:** 3
**Presentation:** 3
**Contribution:** 3
**Rating:** 6
**Confidence:** 4

**Summary:**

This paper introduces a novel benchmark dataset for evaluating the capabilities of multimodal LLMs on diagnosing body-surface conditions. Data has been collected from in-house volunteers and the derived question-answer pairs are reviewed by medical professionals. The paper evaluates state-of-the-art models, including both proprietary and open-source models, and demonstrates that current AI techniques fall far behind senior professionals on diagnosing body-surface conditions.

**Strengths:**

- This work contributes valuable new data collected in-house that can be leveraged to fairly evaluate existing techniques as it is guaranteed that such models are not trained on the dataset.

- The benchmark is generated based on physician annotated data, and all QA pairs are manually reviewed. This provides solid guarantees on its reliability and correctness.

- The paper evaluates human performance on the benchmark as well, thus we can gauge the gap between existing models and medical professionals.

**Weaknesses:**

- It is unclear if the physicians participating in annotation and QA verification overlap with the medical professionals asked to solve the benchmark for human-performance baseline. If there is an overlap, then I am doubtful about the methodology as the human performance may be inflated.

- The part about automatic scoring needs more clarification. For multiple-choice evaluation, why cannot models simply output the selected answer option, which is then compared to the ground truth?

- It would be great to have more analysis on *how* the models fail on the benchmark, leading to a discussion on how to improve current models to close the gap with senior medical professionals.

Minor:
- Typos on lines 198, 392.
- The Appendix is missing.

**Questions:**

- Has there been an overlap between the experts creating the dataset and those solving the benchmark for the reported numbers?
- Have authors observed consistent failure modes on the benchmark? If so, what is the take-away in terms of future directions?

---

> ### Author Response · Authors · 2025-11-21
> **Response to Reviewer PsWv**
>
> Dear Reviewer PsWv,
>
> We would like to express our sincere gratitude to the reviewers for their meticulous review and valuable feedback on our manuscript. We highly appreciate the recognition of the key strengths of our work, which further affirm the significance and contribution of our research. Below, we address each of the raised concerns in detail and provide clarifications to supplement and improve our paper.
>
> **W1: Relationship between the experts creating the dataset and those solving the benchmark for the reported numbers**
>
> A1: Thank you for the comment. Regarding the concern about potential overlap between the experts involved in dataset construction and those involved in solving the benchmark for the reported results, we confirm that no such overlap exists. Our annotation workflow involves two independent groups: executive annotators and review annotators. Executive annotators performed the annotations strictly according to the predefined standards, whereas review annotators focused on identifying and removing problematic samples.
>
> For the human benchmark performance reported in the paper, the results were obtained from a separate group of physicians with relevant clinical specializations. These physicians completed the benchmark tasks independently and had no participation in any stage of the dataset creation process.
>
> **W2: For multiple-choice evaluation, why cannot models simply output the selected answer option, which is then compared to the ground truth?**
>
> A2: Thank you for the comment. We would like to clarify that our evaluation strictly follows this straightforward core rule. In practice, different Large Language Models (LLMs) exhibit varying degrees of instruction-following capability. When responding to multiple-choice questions, some models may generate content beyond the target answer options (e.g., so-called "chain-of-thought" reasoning or explanatory text) rather than directly outputting the selected option(s)[1]. To address this inconsistency, a preliminary step of option extraction is necessary: we first extract the explicit answer option(s) from the model’s full response. Once the option(s) are extracted, we compare them directly with the ground truth for evaluation, following clear scoring criteria:
>
> * No points are awarded for incorrect answers;
> * Partial points are granted for multiple-choice questions with missing correct options (proportional to the number of correctly selected options);
> * Full points are awarded only when the extracted answer exactly matches the ground truth.
>
>
> **W3: Have authors observed consistent failure modes on the benchmark? If so, what is the take-away in terms of future directions?**
>
> A3: Thank you for the comment. Using GPT-4V as an example, we categorized the types of incorrect responses, with detailed analyses provided in Sections B.2 and B.3 of the supplementary materials. In summary, the model’s primary weaknesses fall into three categories: text-understanding deficits, tendencies to refuse to answer, and perceptual misidentification. Specifically:
> - Text understanding: The model at times fails to fully comprehend the question, resulting in irrelevant or partially aligned responses.
> - Refusal to answer: The model occasionally declines to provide an answer to the posed question.
> - Perceptual errors: Multimodal misalignment between visual and textual information can lead to implausible descriptions or misinterpretations of real-world scenes, ultimately impairing spatial, physical, or contextual reasoning.
>
> These observations point to two key implications for future research.
> - First, models require stronger medical-domain knowledge, including improved medical text comprehension and more precise alignment between visual content and textual reasoning.
> - Second, response-generation paradigms should be made more robust so that models can produce professional, contextually appropriate answers to the greatest extent possible given the available information.
>
> **Minor: Typos and missing appendix**
>
> Thank you for the comment. We have corrected all these typos and included the supplementary materials in the revision.
>
> [1] Hu Y, Li T, Lu Q, et al. Omnimedvqa: A new large-scale comprehensive evaluation benchmark for medical lvlm[C]//Proceedings of the IEEE/CVF Conference on Computer Vision and Pattern Recognition. 2024: 22170-22183.

---

> ### Author Response · Authors · 2025-11-26
> **Kindly Requesting Your Thoughts on Our Response**
>
> Dear Reviewer,
>
> Thank you very much for your time and valuable feedback on our submission. As we near the end of the rebuttal period, we would appreciate your thoughts on whether our response has sufficiently addressed your primary concerns. If there are any additional suggestions or clarifications, we are more than willing to engage further to improve the paper.

---

### Official Review · Reviewer_UgBg · 2025-10-23

**Soundness:** 3
**Presentation:** 3
**Contribution:** 2
**Rating:** 4
**Confidence:** 4

**Summary:**

The paper introduces MedLesionVQA, a multimodal benchmark aimed at emulating the real clinical visual diagnostic workflow for body-surface health (dermatology, dentistry, surgery). It comprises ~12K in-house volunteer images and ~19K expert-verified QA pairs, annotated across 94 lesion types, 96 diseases, and over 110 body regions, and organized into seven stepwise abilities: lesion recognition, attribute recognition, location recognition, spatial relation, lesion reasoning, disease diagnosis, and suggestion & treatment. The authors propose an automated scoring pipeline for MCQ and open-ended items (with a judge-LLM) and compare over 20 MLLMs against primary and senior physician baselines, showing a persistent performance gap.

**Strengths:**

- The paper provides a comprehensive benchmark of over 20 general-domain MLLMs and physicians over fine-grained body-surface VQA tasks, with concrete findings on the specialized capabilities of these models.
- The dataset is of good quality with manual quality check, covering a broad range of cases, question types, body-region ontologies, and lesion attributes, supporting comprehensive assessment.
- The writing highlights practical insights that can guide future model design and training.

**Weaknesses:**

**1. About the queries solvable by text-only LLM**

I am curious why the authors do not drop VQA items that can be easily answered by a text-only LLM, which indicates that the question itself can be hacked without using the visual evidence and should not be included for VQA benchmarking. This is a common practice to avoid language shortcuts in curating VQA benchmarks [1--3].

[1] MMMU-Pro: A More Robust Multi-discipline Multimodal Understanding Benchmark (ACL 2025)

[2] PMC-VQA: Development of a large-scale medical visual question-answering dataset (Communication Medicine 2024)

[3] MicroVQA: A Multimodal Reasoning Benchmark for Microscopy-Based Scientific Research (CVPR 2025)


**2. About evaluation of medical-domain models**

Currently the benchmark only assesses general-domain MLLMs. It would provide more insights if including medical-domain [4-6] or dermatology-specialized MLLMs [7].

[4] MedGemma Technical Report (arxiv 2025)

[5] HuatuoGPT-vision, towards injecting medical visual knowledge into multimodal llms at scale (EMNLP 2024)

[6] GMAI-VL & GMAI-VL-5.5M: A Large Vision-Language Model and A Comprehensive Multimodal Dataset Towards General Medical AI (arxiv 2024)

[7] Pre-trained Multimodal Large Language Model Enhances Dermatological Diagnosis using SkinGPT-4 (Nature Communications, 2024)


**3. About more details**
- The paper notes large gaps in visually demanding tasks. I am wondering if there are per-ability error taxonomies for physicians versus models. That would sharpen which capabilities (e.g., boundary/size/color perception vs. region localization) most need architectural or training changes.
- The authors mention that they use prompt tweaks for the open-ended scoring based on GPT-4 after observing disagreements (e.g., color/size strictness). This introduces circularity and instability: results hinge on a proprietary model’s behavior and prompt details. There seems no rigorous inter-rater reliability report (e.g., Cohen’s κ) between physicians and the judge-LLM. It seems that the authors did not upload the appendix; there is no content at the end of the PDF nor supplementary material.


**4. Discussion with highly related works is missing**. For example, DermVQA [8] and Derm1M [9].

[8] DermaVQA: A Multilingual Visual Question Answering Dataset for Dermatology (MICCAI 2024)

[9] Derm1M: A Million‑Scale Vision‑Language Dataset Aligned with Clinical Ontology Knowledge for Dermatology (ICCV 2025)


**5. Question on the LLM-based scoring**

The use of commercial proprietary LLMs (GPT-4) might entail model/version drift risk since neither the API nor any specific snapshot is guaranteed to be available forever. Given this, I am wondering whether open-source models (e.g. Qwen series) are able to serve this and how the evaluation results will vary (e.g., will the scores differ drastically, or will there be any bias?).

---

Minor issues:
- The “multiple-choice question” (selecting one single answer from multiple choices) in Line 277 seems to be “multiple-response question” (selecting all correct answers).
- The first sentence in the Introduction section is a strong assertion (“Taking a photo and consulting multimodal large language models has become a main approach for addressing body surface health concerns”). I am not sure if this is the case for the global world. Any references/statistics to support such solution as “a main approach”?
- Most of the references in this manuscript should use “\citep” with parentheses to avoid confusion with the main texts.
- In Figure 1, upper right part, “Leision Recognition” → “Lesion Recognition”.

**Questions:**

Will the dataset be made publicly available?

---

> ### Author Response · Authors · 2025-11-21
> **Response to Reviewer UgBg [1/3]**
>
> Dear Reviewer UgBg,
> Thank you for your constructive feedback and for pointing out the missing appendix, which is caused by a submission/upload issue. We have fixed it in the revised version.
>
> **W1: About the queries solvable by text-only LLM**
>
> A1: Thank you for your thoughtful feedback. We plan to release two versions of the benchmark: 1) a comprehensive version that includes all questions, and 2) a subset where questions cannot be answered by text-only models.
> However, we would like to clarify that there are rare questions solvable by text-only LLM except for text-highly-related questions, like  suggestions and treatment questions. GPT4V has a score of 0.2201 with text-only inputs, while the score increases to 0.4071 by using both image and text inputs for the Lesion recognition. The reason why keeping these questions in benchmarks is as follows:
> Our primary goal is to emulate the real-world clinical diagnostic process, where both visual and textual information are integrated. To achieve this, we intentionally retained all questions, reflecting the stepwise diagnostic reasoning used by clinicians. For example, questions about suggestions and treatments often rely more on text than on visual data. Removing text-related questions might skew the benchmark, causing it to over-rely on visual input and not fully represent the clinical context where both modalities are used.
> We believe that including these queries ensures a more holistic evaluation of multimodal models' performance. We are open to further refinements in future versions of the benchmark, and your suggestion will certainly help guide this process.
>
> **W2: About evaluation of medical-domain models**
>
> A2: Thank you for providing these highly relevant references and for the excellent suggestion to include more medical-domain and dermatology-specialized MLLMs in our evaluation. We agree that this is crucial for comprehensive analysis.
>
> In response, we now evaluate the models for which we can obtain publicly available weights, including MedGemma, HuatuoGPT, Biomedgpt, Bimedix2, LLava-med, lingshu7b, and lingshu32b.  As shown in the table below, these models exhibit suboptimal performance. We attribute this to two primary factors:
> - Reliance on supervised fine-tuning (SFT) for domain adaptation. Although SFT is effective, it often leads to overfitting to domain-specific patterns (e.g., radiology report generation), which can constrain a model’s ability to generalize to unseen tasks—such as our body-surface–focused benchmark—and may even degrade underlying reasoning capabilities.
> - Limitations of model architecture. Recent foundational MLLMs possess substantially stronger general-purpose visual capabilities. Models built upon less capable architectures therefore start with a lower performance ceiling, making adaptation to complex, out-of-domain tasks even more difficult regardless of the fine-tuning strategy employed.
>
> | Model              | Avg    | lesion recognition | region recognition | attribute recognition | spatial relation | lesion reasoning | disease diagnosis | suggestion treatment |
> |--------------------|--------|---------------------|---------------------|------------------------|-------------------|-------------------|--------------------|-----------------------|
> | Biomedgpt-instruct | 0.0025 | 0.0009              | 0.002               | 0.0008                 | 0.0115            | 0.0016            | 0.0035             | 0.0055                |
> | LLava-Med          | 0.0791 | 0.0372              | 0.0715              | 0.1104                 | 0.1258            | 0.0466            | 0.0535             | 0.1426                |
> | MedGemma-4b        | 0.203  | 0.1212              | 0.0915              | 0.1697                 | 0.4007            | 0.1908            | 0.2339             | 0.5564                |
> | Biomedix2          | 0.2676 | 0.171               | 0.1647              | 0.3118                 | 0.4095            | 0.1563            | 0.349              | 0.5977                |
> | Huatuo-GPT         | 0.279  | 0.2281              | 0.1111              | 0.2997                 | 0.5649            | 0.1897            | 0.282              | 0.676                 |
> | Lingshu7b          | 0.3448 | 0.2058              | 0.271               | 0.379                  | 0.504             | 0.2822            | 0.2794             | 0.7078                |
> | Lingshu32b         | 0.3539 | 0.21                | 0.2723              | 0.3796                 | 0.5551            | 0.30              | 0.2811             | 0.7276                |
>
> While GMAI-VL and SkinGPT-4 are considered for our evaluation, they are currently omitted. GMAI-VL is excluded due to the unavailability of its public model weights. The integration of SkinGPT-4 poses technical and time constraints that prevent its inclusion. We intend to incorporate these models in future iterations of our benchmark upon their availability and successful integration.

---

> ### Author Response · Authors · 2025-11-21
> **Response to Reviewer UgBg [2/3]**
>
> **W3-1: More details about per-ability error taxonomies for physicians versus models**
>
>
> |                          | lesion recognition | location recognition | attribute recognition | spatial relation | lesion reasoning | disease diagnosis | suggestion treatment |
> |--------------------------|--------------------|-----------------------|------------------------|-------------------|-------------------|--------------------|------------------------|
> | **Senior physicians**    | 0.7321             | 0.6826                | 0.7583                 | 0.7046            | 0.7102            | 0.6533             | 0.7313                | 0.8574 |
> | **Primary physicians**   | 0.6144             | 0.5932                | 0.6218                 | 0.5203            | 0.6336            | 0.5412             | 0.6258                | 0.8162 |
> | **Gemini-2.5-pro**       | 0.5252             | 0.4902                | 0.5166                 | 0.4300            | 0.6223            | 0.5754             | 0.6048                | 0.8482 |
> | **Senior. vs Gemini-2.5-pro**    | 20.69%             | 19.24%                | 24.17%                 | 27.46%            | 8.79%             | 7.79%              | 12.65%                | 0.92%  |
> | **Primary. vs Gemini-2.5-pro**   | 8.92%              | 10.30%                | 10.52%                 | 9.03%             | 1.13%             | -3.42%             | 2.10%                 | -3.20% |
>
>
> A3-1: We thank the reviewer for this insightful suggestion. We provide a per-ability comparison between physicians and Gemini-2.5-pro, which has  the best performance in MedLesionVQA . Several patterns emerge: (i) physicians consistently outperform Gemini-2.5-pro on visually demanding recognition and reasoning abilities (e.g., lesion and attribute recognition, region localization), especially for location recognition and disease diagnosis; (ii) the gap narrows, and in some cases reverses, on more text-centric abilities such as suggestion & treatment. We believe this quantitative breakdown already sharpens which capabilities most require architectural or training improvements.
> In addition, we report and qualitatively analyze error distributions over broader categories, such as lack of medical knowledge, text misunderstanding, and judgment errors (Appendix B.2–B.3). Extending this into a systematic, per-ability error taxonomy (e.g., boundary/size/color vs. region localization vs. spatial relation) for both physicians and models would require additional targeted annotation and careful study design, which we view as an important but non-trivial direction for future work.
>
> **W3-2: Prompt tweaks to mitigate disagreement and inter-rater reliability between physicians and the judge-LLM**
>
> Thank you for your valuable feedback regarding prompt refinements to mitigate disagreement and the inter-rater reliability analysis.
> We would like to clarify the motivation behind our prompt enhancement strategy. Through examining inconsistent cases, we observed that the judge LLM often applies overly strict criteria for certain attributes (e.g., color and size) due to limited medical background knowledge. For example, blood scab and crust should both map to scab, and shades such as brown, dark brown, and black are typically grouped into a dark-color category in clinical practice.
> Rather than “hacking” the judge LLM through prompt engineering, we incorporated concise medical knowledge context into the default evaluation prompt to guide the model toward more objective and clinically aligned scoring, thereby improving agreement. This added context is effective for both proprietary models (e.g., GPT-4) and open-source models (e.g., Qwen).
> Thank you as well for the suggestion regarding Cohen’s Kappa coefficient. We have conducted a rigorous analysis to quantify the agreement between the judge LLM and a senior physician on the open-ended evaluation tasks.
> In short, the quadratic-weighted Cohen’s Kappa for open-ended questions is $kappa_w = 0.8825$.
> The resulting value (\kappa_w = 0.8825 > 0.8) reflects high consistency between the judge-LLM and the human expert. We will include the full computation details and reliability analysis in the revised Appendix A.5.

---

> ### Author Response · Authors · 2025-11-21
> **Response to Reviewer UgBg [3/3]**
>
> **W4: Discussion with highly related works is missing**
>
> We thank the reviewer for pointing out these highly relevant works. In the final version, we will explicitly discuss DermaVQA and Derm1M in the Related Work section. DermaVQA introduces a multilingual dermatology VQA dataset built from remote-care patient portal messages with user-generated images, focusing on response generation for dermatology consultations.  Derm1M, in contrast, is a million-scale dermatology vision–language dataset aligned with a clinical ontology and primarily designed as a large-scale pretraining resource for CLIP-style models and downstream classification/retrieval tasks.
> Our MedLesionVQA benchmark is complementary: it targets body-surface health across multiple specialties (dermatology, dentistry, surgery, etc.)  and is explicitly constructed as a physician-aligned, workflow-level evaluation benchmark with fine-grained lesion, region, and reasoning abilities rather than as a pretraining corpus. We will clarify these distinctions and the complementary roles of these datasets in the final manuscript.
>
> **W5: Questions on LLM-based scoring**
>
> A5: Thank you for this crucial insight. We agree that the potential for model/version drift in proprietary LLMs like GPT-4 is a significant concern for the reproducibility of research findings. To address this, we conduct a new set of experiments using models from the open-source Qwen 2.5 series (specifically, the 7B, 32B, and 72B parameter models) as alternative judges. We re-evaluated the performance of Huatuo-GPT model on the MedLesion-VQA dataset. The detailed results, comparing the original GPT-4 judge with the Qwen 2.5 series, are presented in the table below:
>
> | Judger                                | Avg | lesion recognition | region recognition | attribute recognition | spatial relation | lesion reasoning | disease diagnosis | suggestion treatment |
> |---------------------------------------|----------|---------------------|---------------------|------------------------|-------------------|-------------------|--------------------|-----------------------|
> | Huatuo-GPT (qwen2.5 7b as judger)     | 0.3516   | 0.2878              | 0.1684              | 0.3615                 | 0.5649            | 0.3033            | 0.3777             | 0.717                 |
> | Huatuo-GPT (qwen2.5 32b as judger)    | 0.3359   | 0.2878              | 0.1684              | 0.3715                 | 0.587             | 0.2424            | 0.2984             | 0.7141                |
> | Huatuo-GPT (qwen2.5 72b as judger)    | 0.3065   | 0.2453              | 0.1292              | 0.3303                 | 0.5768            | 0.2312            | 0.2938             | 0.7158                |
> | Huatuo-GPT (GPT4 as judger)           | 0.279    | 0.2281              | 0.1111              | 0.2997                 | 0.5649            | 0.1897            | 0.282              | 0.676                 |
>
> We observed a clear trend: as the judge model's size increases, its scoring becomes stricter. For instance, Qwen-2.5-72B's scores are lower than the 7B model's and align more closely with GPT-4's. We hypothesize this is not a bias, but a result of increased knowledge depth. Larger models are more discerning of subtle errors, leading to more cautious and human-like evaluations, while smaller models may be more lenient.
>
> ---
>
> **Minor issues:**
> **M1, 3, 4:  Typos and Writings**
>
> Thank you for pointing out the terminology confusion regarding “multiple-response question,” the usage of “\citep,” and the typo in “Lesion Recognition.” In the revised version, we have replaced the phrase “multiple-choice question” with “multiple-response question” for improved clarity.
>
>
> **M2: The first sentence in the Introduction section is a strong assertion**
> We appreciate the need for a more precise qualification of this statement. A review of recent literature indicates growing interest in photo-based workflows combined with multimodal large language models (MLLMs) in dermatology and body-surface health [1, 2]. However, we agree that characterizing these workflows as a main global approach may overstate their current level of adoption.
> Following your suggestion, we propose revising the sentence to:
> “Photo-based interaction with multimodal large language models has recently gained attention as a potential pathway for addressing body-surface health concerns.”
>
> ---
>
> **Q: Will the dataset be made publicly available?**
>
> A: Yes, we will release the dataset and code directly in the camera-ready version.
>
> [1] AlSaad, Rawan, et al. "Multimodal large language models in health care: applications, challenges, and future outlook." Journal of medical Internet research 26 (2024): e59505.
> [2] Zhou, Juexiao, et al. "Pre-trained Multimodal Large Language Model Enhances Dermatological Diagnosis using SkinGPT-4." medRxiv (2023): 2023-06.

---

> ### Author Response · Authors · 2025-11-26
> **Kindly Requesting Your Thoughts on Our Response**
>
> Dear Reviewer,
>
> We sincerely appreciate your thorough feedback on our paper. As the rebuttal period is coming to a close, could you please confirm if our response has adequately addressed your concerns? If you have any remaining suggestions or would like to continue the discussion, we would be more than happy to make further adjustments.

---

> ### Comment · Reviewer_UgBg · 2025-11-27
>
> **1. About W1**
> > "a subset where questions cannot be answered by text-only models"
>
> What is the size/portion of this subset? Could the author provides one or two examples?
>
> > "Questions about suggestions and treatments often rely more on text than on visual data...Removing text-related questions might skew the benchmark, causing it to over-rely on visual input..."
>
> This claim seems plausible. But the paper itself is focusing on **V**QA (as in the title and throughout the paper). A **Visual** Question Answering benchmark **should rely on visual input**. That is not a "skew"; that is the definition of the test.
>
>
> I appreciate the authors for splitting the benchmark to two subsets ("all" + "text-only") and performing further analysis. But given the scope of the paper (at least the content it now presents), I would suggest the authors split it into "multimodal (vision+text)" and "text-only", following similar work [10], instead of framing it as a "VQA benchmark" while arguing "over-relying on visual input should be avoided". No hurry. This is not asking for immediate revision, as it takes time to reorganize the results.
>
> [10] MedXpertQA: Benchmarking expert-level medical reasoning and understanding. (ICML 2025)
>
>
> ---
>
> **2. About W2**
>
> I appreciate the addition of these experiments; they significantly strengthen the paper. This concern is well addressed.
>
> ---
>
> **3. About W3**
>
> I can now see the appendix.
>
> For the first point, I was expecting a performance breakdown to a taxonomy like Tables 3--6 (a first-level breakdown or a even coarser level already provides many insights), so the community can clearly see the problem: *what specific anatomies / diseases /... that current MLLMs fail to solve and what capabilities need to be further improved*. The per-ability comparison provided in the current response is too general (like "attribute recognition" itself does not provide much information. As in the Appendix Line 857, the breakdown of "attribute" into "Lesion attributes contain size, color, shape, distribution, quantity, and boundary" is more valuable).
>
> That said, I understand that there are practical and time limitation. So I will not ask the authors to rush the experiments. Instead, I would like to hear from authors about their ideas on the problem described above, based on their experience.
>
> ---
>
> **4. About W4**
>
> This concern is addressed.
>
> ---
>
> **5. About W5**
>
> I appreciate the additional test on other LLM judges. The results show that larger open-source models (72B) provide stricter scores that align closer to GPT-4. This result provides experimental support for the rationale of using strong open-source LLMs as a judge, and can be included in the revised version.
>
> ---
>
> **Additional questions raised from the appendix**
>
> In Table 11 of the newly uploaded appendix, there is CoT prompting for GPT-4.1 and Gemini-2.5-Pro. Does it mean manually feeding prompts like "Think step by step...", or turning on the "thinking mode"? It is surprising to see that the performance of these models with CoT is no significantly different (or even worse) than zero-shot or one-shot setting. Could the authors explain this?
>
>
> ---
>
> Other minor comments are addressed. I will consider raising the score if the remaining concerns are cleared.

---

> ### Author Response · Authors · 2025-11-29
> **Official Comment by Authors [1/2]**
>
> Dear Reviewer UgBg,
> We sincerely appreciate your insightful comments and the highly positive discussion. Our responses to your concerns are as follows:
>
> **1. About W1**
> > What is the size/portion of this subset? Could the author provide one or two examples?
>
> Following your advice, we calculated that the subset which can be answered without significant reliance on visual input constitutes approximately 9% of the dataset. This was identified through a cross-validation process using both GPT-4V and Qwen2.5-72B to flag such cases.
>
> The cases mainly belong to the category of treatment and suggestion. Here are two representative examples:
> - Question: "Considering the papules that protrude slightly from the skin surface around the mouth in the image, what treatment method and medication should be taken? A. Adapalene gel;  B. Vitamin A acid cream; C. External antibiotics such as fusidic acid cream and erythromycin cream;  D. Doxycycline"
> - Ground Truth (GT): C. External antibiotics such as fusidic acid cream and erythromycin cream
> - Analysis: In this scenario, image information plays a role in identifying the lesion's attributes, including distribution, color, and size, which can indicate the degree and phases of disease development. The model can still answer correctly with a high probability, even without relying heavily on the image clues, because the treatment of perioral inflammation and papules on the skin of the mouth is highly related to "fusidic acid cream and erythromycin cream" in most text croupus.
>
> ---
> - Question: "The patient in the image has been diagnosed with tinea corporis with large and relatively severe lesions. Which oral antifungal drugs are recommended?"
> - Ground Truth (GT): Itraconazole capsules, Fluconazole capsules
> - Analysis: Here, the vision component is responsible for only a minimal part of the final answer. It is used merely to verify a visual attribute (severity), while the core task of recommending specific drugs (e.g., 'Itraconazole') depends entirely on the model's internal, text-based knowledge base.
> ---
>
> > A Visual Question Answering benchmark should rely on visual input. That is not a "skew"; that is the definition of the test.
>
> We agree that the fundamental purpose of VQA is to test a model's understanding and reasoning abilities based on visual input. Our original intention in including questions about suggestions and treatment—which rely on the model's text-based medical knowledge—was to gauge its potential as a practical, comprehensive clinical tool. However, these text-heavy questions dilute the definition of a VQA task. Therefore, we will separate them out in the released version of our benchmark.
>
> >  I would suggest the authors split it into "multimodal (vision+text)" and "text-only".
>
> Thank you for this highly constructive feedback. We agree that splitting the dataset into "multimodal (vision+text)" and "text-only" is a more accurate way. Our initial intention with the "all" vs. "text-only" split was to measure the "net performance gain" from adding the visual modality. We appreciate that the reviewer has correctly pointed out a logical inconsistency between the 'VQA' framing and our argument against an over-reliance on visual input. We also sincerely appreciate your kindly consideration in noting that a full re-organization of the results would be time-intensive. We will adopt the "multimodal" vs. "text-only" categorization for the public release of the dataset.

---

> ### Author Response · Authors · 2025-11-29
> **Official Comment by Authors [2/2]**
>
> **2. About W3**
> Thank you for the constructive suggestion. In response, we conduct a more granular analysis of the "attribute recognition" capability to provide more specific insights. We perform a fine-grained evaluation on the open-ended "attribute recognition" questions from our validation set. This assessment measured the performance of several leading MLLMs (e.g., Qwen-2.5VL-72B, GPT-4V, Gemini 2.5 Pro) across six key sub-attributes. The aggregated results are presented in the table below.
>
> | Model              | Attribute-Distribution | Attribute-Size | Attribute-Shape | Attribute-Count | Attribute-Boundary | Attribute-Color |
> |--------------------|------------------------|----------------|-----------------|-----------------|--------------------|-----------------|
> | Qwen2.5-VL-72B-    | 0.3604                 | 0.2361         | 0.2793          | 0.6723          | 0.578              | 0.4392          |
> | GPT-4V              | 0.3937                 | 0.4166         | 0.3322          | 0.7243          | 0.6716              | 0.4435          |
> | Gemini 2.5 Pro     | 0.4802                 | 0.3391         | 0.3115          | 0.6847          | 0.5624              | 0.4957          |
> | AVG.               | 0.4114                 | 0.3306         | 0.3077          | 0.6938          | 0.604               | 0.4595          |
>
> The results reveal a stark disparity in performance across these attributes. While the models demonstrate comparatively better performance on simpler, more quantifiable attributes such as lesion Count and Boundary, they struggle significantly with attributes that demand more nuanced perception and description, like Shape, Size, and Distribution.
>
> These findings collectively demonstrate that current MLLMs exhibit significant shortcomings in the fine-grained perception and fine-grained description of skin lesions. We will incorporate this insightful discussion into our revised appendix.
>
> **3. Additional questions raised from the appendix**
> Thank you for this insightful question. We also noticed this unusual finding. First,  we implemented CoT via standard prompting (Let's think step by step...) and did not activate a special model mode. Second, we attribute the observed performance degradation to two primary factors:
> 1. CoT is most effective for tasks requiring explicit, multi-step logical deduction (e.g., mathematics, symbolic reasoning). In contrast, the bottleneck in our VQA tasks is not complex reasoning but initial visual perception—an often instantaneous and holistic process. Applying a sequential reasoning framework to a perception-centric task is fundamentally inefficient and can be ineffective.
> 2. For large closed-source models (e.g., Gemini 2.5 Pro), their internal mechanisms are opaque when accessed via an API. One possibility is that these models already integrate efficient reasoning capabilities into their inference processes—when prompted for CoT, they do not reveal a real-time thought process but rather perform post-hoc rationalization: generating plausible-sounding steps to justify a conclusion already reached internally.
>
>
> Once again, we would like to express our sincere gratitude for your detailed and constructive feedback. Your suggestions have been invaluable in helping us refine and improve our work. We hope that our responses and the additional analyses have adequately addressed your concerns, and we would be happy to provide further clarification on any remaining points.

---

### Official Review · Reviewer_ayvc · 2025-10-27

**Soundness:** 3
**Presentation:** 2
**Contribution:** 2
**Rating:** 4
**Confidence:** 4

**Summary:**

This paper introduces MedLesionVQA, a multimodal benchmark designed to test how well AI systems can perform medical visual diagnosis for body surface conditions such as skin, oral, and nail diseases. It includes over 12k volunteer images and VQA pairs that cover seven diagnostic steps including lesion recognition to treatment recommendation. The dataset is curated with annotations across several lesion types, diseases, and body regions, reflecting real clinical workflows validated by clinical experts. The authors evaluate around 20 MLLMs and find that the best model achieves 56.2% accuracy, still well below physician performance. This work provides interesting insights evaluating MLLMs for body surface diagnosis.

**Strengths:**

- The authors experimentally show limitations of current MLLMs by comparing them against human physicians across realistic diagnostic tasks, providing quantitative evidence that even the best models are behind expert clinicians.
- The paper offers valuable empirical analysis on scaling and specialization trade-offs, showing through systematic evaluation that while larger models generally perform better, domain specific fine-tuning can sometimes reduce performance.

**Weaknesses:**

- The paper does not disclose information about the diversity of participants involved in the data collection process. Metrics such as the country of origin and age groups of participants would help assess whether the benchmark is truly diverse or if its biased to any particular demographic or target group.
- Need to include comparisons about other Medical MLLMs like – Google’s MedGemma [1], Huatuogpt [2], Bimedix2 [3]. This would help us understand if domain specific medical instruction tuning affects the model’s generalizability and how these models compare on MedLesionVQA.
- The Appendix and the supplementary material section are missing from the submission. The authors make multiple references to it in the main paper.  Detailed prompts for the evaluation framework are missing.
- In Figure 1 Suggestion and Treatment section “list at least two topical anti-infective drugs?”. Questions like these would help models respond without properly analyzing the image. This reduces the credibility of the benchmark.

[1] *Sellergren, Andrew, et al. "Medgemma technical report." arXiv preprint arXiv:2507.05201 (2025).*

[2] *Chen, Junying, et al. "Huatuogpt-vision, towards injecting medical visual knowledge into multimodal llms at scale." arXiv preprint arXiv:2406.19280 (2024).*

[3] *Mullappilly, Sahal Shaji, et al. "Bimedix2: Bio-medical expert lmm for diverse medical modalities." arXiv preprint arXiv:2412.07769 (2024).*

**Questions:**

Please address the above weaknesses.
- What instructions were given to the clinical experts for verification of the samples?
- Please include the clinical lexicon tree in the appendix.

---

> ### Author Response · Authors · 2025-11-21
> **Response to Reviewer ayvc [1/2]**
>
> Dear Reviewer ayvc,
>
> Thanks for your serious and thoughtful comments. We apologize for the omission of the appendix in the submission. The appendix is updated in the revised version, including the clinical lexicon tree and detailed prompts. For other concerns, we will reply to your comments as follows:
>
> **W1: Information about the diversity of participants**
>
> A1: We recruit volunteers aged 15 to 75 years old, with the highest frequency in the 40- to 45-year-old age group, the information is also mentioned in line 198-200 in the submission. Specifically, Considering the privacy protection of minors in ethical approvals, children younger than 15 years old are left out of data collection. Details of age and gender statistics can be found in appendix A.1.
>
> **W2: Evaluation results of more medical MLLMs**
>
> A2: Thank you for this valuable suggestion. We benchmarked several medical MLLMs (e.g., MedGemma, HuatuoGPT, Bimedix2) on MedLesionVQA. As shown in the table below, these models exhibit suboptimal performance. We attribute this to two primary factors:
> - Reliance on supervised fine-tuning (SFT) for domain adaptation. Although SFT is effective, it often leads to overfitting to domain-specific patterns (e.g., radiology report generation), which can constrain a model’s ability to generalize to unseen tasks—such as our body-surface–focused benchmark—and may even degrade underlying reasoning capabilities.
> - Limitations of model architecture. Recent foundational MLLMs possess substantially stronger general-purpose visual capabilities. Models built upon less capable architectures therefore start with a lower performance ceiling, making adaptation to complex, out-of-domain tasks even more difficult regardless of the fine-tuning strategy employed.
>
> Thank you again for the helpful suggestion. We will incorporate this comparative analysis into the final manuscript.
>
> | Model              | Avg    | lesion recognition | region recognition | attribute recognition | spatial relation | lesion reasoning | disease diagnosis | suggestion treatment |
> |--------------------|--------|---------------------|---------------------|------------------------|-------------------|-------------------|--------------------|-----------------------|
> | Biomedgpt-instruct | 0.0025 | 0.0009              | 0.002               | 0.0008                 | 0.0115            | 0.0016            | 0.0035             | 0.0055                |
> | LLava-Med          | 0.0791 | 0.0372              | 0.0715              | 0.1104                 | 0.1258            | 0.0466            | 0.0535             | 0.1426                |
> | MedGemma-4b        | 0.203  | 0.1212              | 0.0915              | 0.1697                 | 0.4007            | 0.1908            | 0.2339             | 0.5564                |
> | Biomedix2          | 0.2676 | 0.171               | 0.1647              | 0.3118                 | 0.4095            | 0.1563            | 0.349              | 0.5977                |
> | Huatuo-GPT         | 0.279  | 0.2281              | 0.1111              | 0.2997                 | 0.5649            | 0.1897            | 0.282              | 0.676                 |
> | Lingshu7b          | 0.3448 | 0.2058              | 0.271               | 0.379                  | 0.504             | 0.2822            | 0.2794             | 0.7078                |
> | Lingshu32b         | 0.3539 | 0.21                | 0.2723              | 0.3796                 | 0.5551            | 0.30              | 0.2811             | 0.7276                |
>
>
>
> **W3-1: Appendix and the supplementary material section are missing from the submission**
> A3-1:  We are sorry for mistakenly dropping the appendix. We have uploaded the appendix in the revised version.
>
> **W3-2: Detailed prompts for the evaluation framework**
> A3-2: We illustrate the evaluation pipeline in Sec 3.5, and detailed prompts for multi-choice question and open-ended question evaluation are shown in Fig. 3(b) in the submission. The QAs related with lesion recognition and diagnosis reasoning are  shown in Fig. 3(a). Moreover, other prompts for QA construction can be found in Appendix A.3. We hope the clarification will solve your concerns.

---

> ### Author Response · Authors · 2025-11-21
> **Response to Reviewer ayvc [2/2]**
>
> **W4: The constraints in open-ended questions**
>
> A4: Thanks for the comment. We would like to clarify that the phrase “list at least two topical anti-infective drugs” is used as a constraint suffix, rather than as a standalone question. Such constraints are appended to certain open-ended questions to make the assessment objectives more explicit. Without these constraints, MLLMs often generate diverse or overly general responses, which can reduce the directionality of the assessment and complicate the automatic scoring pipeline. Importantly, all constraints are carefully designed and reviewed to prevent shortcut exploitation without genuine image analysis.
>
> For example, consider the question “How to treat the disease shown in the picture? List at least two topical anti-infective drugs.” If the constraint “list at least two topical anti-infective drugs” is removed, MLLMs may propose alternative management strategies—such as oral medications, adjuvant therapies, or other topical antifungal agents (e.g., bifonazole, miconazole, terbinafine solution). Moreover, topical anti-infective drugs constitute a broad category, not limited to Mopirocin ointment and fusidic acid ointment. The model is expected to first recognize the disease (Tinea pedis) from the image and then provide appropriate examples—such as mupirocin ointment and fusidic acid ointment—which belong to this drug class and are suitable for treating Tinea pedis.
>
> We hope this explanation addresses your concerns and we welcome any further feedback.
>
> **Q1: Instructions for clinical expert verification**
>
> A1: Following your suggestion, we are currently undergoing the protocol for releasing the annotation details, which were generated by senior physicians after multiple rounds of discussion. The preliminary procedures are as follows.
> The core objective of expert verification is criteria alignment. A panel of five clinical experts establishes the annotation standards and refines the guidelines through representative cases. After the completion of data annotation, the same five experts review all annotated samples and correct any errors. The final, revised annotations can therefore be considered criterion-aligned based on their collective assessment.
>
> **Q2: Clinical lexicon tree in the appendix.**
>
> A2: Thank you for pointing this out. Please refer to Appendix A.2 in the revised version.

---

> ### Author Response · Authors · 2025-11-26
> **Kindly Requesting Your Thoughts on Our Response**
>
> Dear Reviewer,
>
> Thank you again for your insightful comments on our paper. With the rebuttal deadline approaching, we would be grateful if you could let us know whether our response has addressed your main points. Should you have any additional feedback or require further revisions, we are eager to engage in any necessary improvements.

---

### Official Review · Reviewer_noDT · 2025-11-01

**Soundness:** 3
**Presentation:** 3
**Contribution:** 3
**Rating:** 6
**Confidence:** 2

**Summary:**

This paper introduces MedLesionVQA, a multimodal benchmark explicitly designed to evaluate multimodal large language models (MLLMs) in body-surface health diagnosis. Unlike existing datasets focusing on isolated disease classification or lesion recognition, MedLesionVQA covers  seven clinical ability dimensions.
The dataset includes 12K ethically collected volunteer images and 19K expert-verified question–answer pairs, covering 94 lesion types, 96 diseases, and 110 body regions. Over 20 MLLMs, including GPT-5, Gemini 2.5 Pro, and Qwen2.5-VL-72B, are evaluated, alongside human baselines. Results show that the best-performing MLLM (Gemini 2.5 Pro) achieves 56.2% accuracy, lagging behind primary physicians (61.4%) and senior physicians (73.2%), indicating persistent gaps between current MLLMs and clinical reasoning capabilities.

**Strengths:**

The benchmark is well-motivated and methodologically thorough, with a clear focus on aligning AI evaluation with authentic clinical workflows. The data collection process is commendably rigorous: recruiting real volunteers under ethical review and performing multi-stage expert validation with high inter-annotator reliability. The seven diagnostic categories mirror the stepwise reasoning of real physicians, offering a structured and interpretable assessment framework. The inclusion of both open- and closed-source MLLMs, together with human baselines, provides a well-rounded evaluation. The paper is clearly written, integrates comprehensive error analysis, and successfully highlights key challenges in general-purpose MLLMs’ ability to handle fine-grained lesion understanding and multimodal reasoning.

**Weaknesses:**

The benchmark’s contribution primarily lies in data curation and clinical alignment rather than introducing a new evaluation paradigm.
The evaluation setup, though comprehensive, leans heavily on performance reporting without deeper interpretability or diagnostic insights. For instance, while the authors provide category-wise breakdowns, the analysis stops short of identifying failure modes beyond recognition vs. reasoning. The results are descriptive but not analytical: there is little exploration of why models fail (e.g., domain shift, visual ambiguity, prompt misalignment) or how these findings could inform model development.

**Questions:**

See above.

---

> ### Author Response · Authors · 2025-11-21
> **Response to Reviewer noDT**
>
> Dear Reviewer noDT,
>
> We sincerely thank you for the thoughtful and constructive feedback. We are glad that you found the benchmark well-motivated, clinically grounded, and methodologically rigorous, especially regarding our real-volunteer data collection, multi-stage expert verification, and alignment with the authentic diagnostic workflow.
>
> Thank you for your detailed and constructive feedback. We address your concerns below.
>
> **W1: The benchmark’s contribution primarily lies in data curation and clinical alignment**
>
> A1: We agree that high-quality, clinically aligned curation is a central part of our work, but we would like to clarify that the evaluation paradigm introduced in MedLesionVQA is new because we can combine the results together for understanding instead of distinct evaluation like existing medical VQA datasets。
>
> Rather than a flat collection of questions, MedLesionVQA is explicitly organized along a stepwise visual diagnostic workflow with seven abilities. This decomposition is derived from clinical textbooks and expert panels, as illustrated in line 88-89 of the manuscript. The benchmark therefore operationalizes “being like a clinician” as a structured multi-ability evaluation, rather than a single accuracy number on disease labels.
>
>
> **W2: The evaluation setup, though comprehensive, leans heavily on performance reporting without deeper interpretability or diagnostic insights**
>
> A2: Thanks for the valuable suggestion. We clarify that our analysis already provides a structured breakdown of model failure modes (lines 406-407), and we have made this more explicit in the supplementary material. Specifically, our error analysis (Appendix B.3, lines 1162–1181) identifies three major categories:
>
> * Text Misunderstanding — limitations in semantic parsing beyond visual perception, where models misinterpret the question intent and generate irrelevant or partially correct answers.
>
> * Prompt Misalignment or Refusal — cases where models decline to answer or output unrelated content, reflecting difficulties in handling complex or multi-step clinical instructions.
>
> * Domain Shift Due to Limited Knowledge — further decomposed into Category Confusion (misclassification of lesion types), Perceptual Error (incorrect spatial or attribute interpretation), and Terminology Precision (use of vague or incorrect medical terms).
> These categories offer a concrete explanation of failure patterns across the seven diagnostic abilities and we hope it can directly address your concerns.
>
> Building on these identified failure patterns, we also clarify how our findings can inform future model development.
>
> - First, the Text Misunderstanding category highlights the need for improved medical instruction–semantics alignment, suggesting that models would benefit from enhanced and diverse clinical question queries.
>
> - Second, addressing Prompt Misalignment or Refusal requires more robust grounding of clinical workflows in model training, including better handling of multi-step instructions.
>
> - Finally, the Domain Shift errors indicate the need for strengthening medical visual understanding: enriching training corpora with high-quality and fine-grained lesion image taxonomies, improving the multimodal perception alignment, and incorporating medical terminology corpora to reduce vague or incorrect outputs.
>
> Together, these directions illustrate how our error taxonomy not only explains existing limitations but also provides actionable guidance for building next-generation medical foundation models with stronger reliability and clinical alignment.

---

> ### Author Response · Authors · 2025-11-26
> **Kindly Requesting Your Thoughts on Our Response**
>
> Dear Reviewer,
>
> We truly appreciate the time and effort you've put into reviewing our paper. As the rebuttal period is nearing its close, we wanted to kindly ask if our response has addressed your key concerns. If you have any additional suggestions or would like further clarification, we are more than happy to continue the discussion and make any necessary revisions.

---

### Author Response · Authors · 2025-11-21
**Add appendix into the revised version**

Dear Reviews and Chairs,

We apologize for the omission of the appendix in the initial submission, and we have included the appendix into the revised version (https://openreview.net/pdf?id=BYtqk6AVuL). Besides the appendix, we have corrected all these typos and thanks for the review's effort to point out these typos.

---

### Public Comment · ~Deli_Yu1 · 2026-05-31
**Request for Checking Updated Camera-Ready PDF Compliance for Paper 15645**

Dear PC,

We are writing regarding the camera-ready version of our paper, Paper 15645, “MedLesionVQA: A Multimodal Benchmark Emulating Clinical Visual Diagnosis for Body Surface Health.”

The previous version was flagged for the following issue:
Wrong margins, e.g., decreasing margins on the left and right.

We sincerely apologize for this issue. We have corrected the margins and uploaded an updated version. However, since we do not have access to the official offline checking tool, we are concerned about whether the updated PDF will pass the ICLR format check.

We would be very grateful if you could help check the updated version. If any margin issue remains, could you please let us know which page is affected? We will revise it immediately.

Sincerely,
Paper 15645

---

### Meta-Review · Area_Chair_fpSH · 2026-01-01

**Summary:**

This paper introduces MedLesionVQA, a multimodal benchmark designed to evaluate MLLMs on body-surface health diagnosis tasks. The benchmark comprises ~12K volunteer images and ~19K expert-verified QA pairs, covering 94 lesion types, 96 diseases, and 110 body regions, organized into seven clinical diagnostic abilities. The authors evaluate over 20 MLLMs against human physician baselines, revealing a persistent performance gap between current models and clinical experts.

The reviewers raised several concerns during the initial review phase: (1) the absence of medical-domain specialized models in the evaluation (ayvc, UgBg); (2) the inclusion of questions that can be answered without visual input, which undermines the VQA benchmark definition (ayvc, UgBg); (3) the reliability and reproducibility of the LLM-based automatic scoring pipeline (UgBg, PsWv); (4) the missing appendix and supplementary materials (all reviewers); (5) insufficient fine-grained failure mode analysis (noDT, PsWv, UgBg); and (6) concerns about potential overlap between annotators and human baseline participants (PsWv).

On the positive side, all reviewers acknowledged the rigorous data collection process, the clinical alignment of the benchmark design, and the value of comparing MLLMs against human physicians. The dataset quality and the practical insights for future model development were consistently praised.

**Reviewer Concerns:**

Concerns Adequately Addressed:
1. Medical-domain model evaluation (ayvc, UgBg): The authors conducted comprehensive experiments on 7 additional medical-specific models (BiomedGPT, LLava-Med, MedGemma, Biomedix2, HuatuoGPT, Lingshu-7b, Lingshu-32b). The results demonstrate that general-domain foundation models currently outperform specialized medical models on this benchmark, attributed to stronger visual backbones and the overfitting tendencies of domain-specific fine-tuning. Reviewer UgBg explicitly acknowledged: "This concern is well addressed."

2. Text-only solvable questions (ayvc, UgBg): The authors performed a thorough analysis identifying that approximately 9% of queries (primarily treatment suggestions) can be answered without significant visual input. They provided concrete examples and, importantly, accepted the reviewers' criticism by committing to split the released benchmark into "Multimodal (Vision+Text)" and "Text-only" subsets. This structural change directly addresses the fundamental concern about VQA benchmark definition. Reviewer UgBg appreciated this response and noted: "No hurry. This is not asking for immediate revision."

3. Scoring reliability and reproducibility (UgBg, PsWv): The authors validated their scoring pipeline using open-source Qwen-2.5 series models (7B, 32B, 72B), demonstrating that larger models produce stricter scores aligned with GPT-4. Critically, they computed Cohen's Kappa (κ = 0.8825) between the judge-LLM and senior physicians, indicating high inter-rater reliability. This quantitative evidence substantially strengthens the credibility of the automated evaluation.

4. Missing appendix (all reviewers): The authors acknowledged the submission error and uploaded the complete appendix in the revised version, including the clinical lexicon tree, detailed prompts, and  demographic information.

5. Fine-grained analysis (noDT, PsWv, UgBg): The authors added granular performance breakdowns across six lesion attributes (Distribution, Size, Shape, Count, Boundary, Color) and provided per-ability comparisons between physicians and models. The analysis reveals that models perform relatively well on quantifiable attributes (Count) but struggle with nuanced descriptive tasks (Shape, Size), offering actionable insights for future improvements.

6. Annotator-evaluator overlap (PsWv): The authors clearly confirmed that annotation experts and human baseline physicians are completely independent groups with no overlap, ensuring the fairness of the human performance comparison.

Minor Outstanding Concerns:
1. Demographic diversity: While the authors provided age distribution (15-75 years) and gender statistics, the racial/ethnic composition of the volunteer pool was not fully disclosed. For a skin-focused dataset, this information would strengthen claims about benchmark generalizability.

2. CoT performance degradation: The explanation for why Chain-of-Thought prompting does not improve (and sometimes degrades) performance remains somewhat speculative, though the observation itself is valuable. This is because the bottleneck in the VQA tasks is not complex reasoning but initial visual perception.

These remaining concerns are relatively minor and do not undermine the core contributions of the work.

**Reviewer Scores:**

Reviewer noDT (Original: 6): Likely to remain at 6. The reviewer's main concern about the lack of deeper interpretability was partially addressed through the added fine-grained attribute analysis and error taxonomy.

Reviewer ayvc (Original: 4): Likely to increase to 6. The two major concerns—missing appendix and lack of medical model comparisons—were directly addressed with substantial new experiments. The demographic information was also clarified in the revised appendix.

Reviewer UgBg (Original: 4): Likely to increase to 6. This reviewer engaged most actively in the discussion and explicitly stated: "I will consider raising the score if the remaining concerns are cleared." The authors addressed all major points: medical model evaluation ("This concern is well addressed"), scoring reliability (Cohen's κ provided), and the text-only question issue (commitment to dataset splitting). The reviewer's tone in later exchanges was constructive and appreciative.

Reviewer PsWv (Original: 6): Likely to remain at 6. The concerns about annotator overlap and scoring clarification were adequately addressed. The reviewer did not raise further objections during the discussion.

---

### Decision · Program_Chairs · 2026-01-26

Accept (Poster)